# Direct detection of an NH-π hydrogen bond in an intrinsically disordered peptide

Luigi Russo [1,10], Dipendu Dhar[2,10], Robin Backer [3], Om Prakash[4], Fatima Matroodi[5], Kerstin Overkamp[6], Karin Giller[6], Stefan Becker [6], Christian Griesinger [6], Dieter Willbold[3,7], Barbara Rossi [5,8], Mehdi D. Davari [2] ✉ & Nasrollah Rezaei-Ghaleh [3,7,9] ✉

Hydrogen bonds play crucial roles in functional biomolecular dynamics. It is suggested that non-conventional hydrogen bonds engaging π electrons are prevalent in proteins. The experimental support for their existence is however limited. Here, we provide direct NMR spectroscopic evidence for the existence of an NH-π interaction in an intrinsically disordered peptide (E22G-Aβ40). In particular, we demonstrate the correlation between the amide proton of a glycine residue (Gly22) and the aromatic carbons of its preceding Phe20 through π hydrogen bond-mediated scalar coupling between them, as predicted by density functional theory calculations. Our results present a proof-of-principle example of NH-π interactions in an intrinsically disordered protein (IDP) and suggest the potential prevalence of π hydrogen bonds on the surface of IDPs. Direct experimental verification of NH-π interactions in folded proteins remains for future studies.

Covalent bonds determine the primary structure of proteins, while non-covalent interactions (NCIs) govern their secondary and tertiary structures as well as their stability[1,2]. Furthermore, NCIs are crucial for the stability of intermolecular protein-protein, protein-nucleic acid, and protein-small molecules interactions. Importantly, the dynamics of NCIs play a key role in the functional motions of folded proteins and their complexes. In addition, the NCIs underlie the residual structure, functional dynamics, and interactions of a large class of proteins, namely, intrinsically disordered proteins (IDPs)[3,4].

Hydrogen bonds are an important class of interactions with a crucial role in the structure and function of biomolecules[5]. They are defined as attractive interactions between a donor XH group of a molecule and an acceptor atom (or groups of atoms) Y in another or the same molecule[6]. Classically, the X and/or Y are atoms such as O, N,

and S, which are more electronegative than H, with a dominant energetic contribution provided by the electrostatic term. The NMR-based detection of scalar ($J$) couplings between the donor X (or H) and acceptor Y atoms have unequivocally confirmed the existence of such classical hydrogen bonds[7–9]. A rather unconventional type of weaker hydrogen bonds is formed between the XH (X: O, N, S, or even C) groups as the donor and the delocalized $sp^2$-hybridized π electron system of aromatic rings as the acceptor (Y)[10]. In simple organic systems such as benzene and methane (for CH-π), ammonia (for NH-π), and water (for OH-π), the XH-π hydrogen bonds are predicted to have interaction energies between 1 and 4 kcal.mol$^{-1}$ dominated by the dispersion term in the case of CH-π and the electrostatic term in NH-π and OH-π cases[11,12]. Surveys of protein structures suggest that around 10% of aromatic rings in the compact cores of folded proteins are

[1]Department of Environmental, Biological and Pharmaceutical Science and Technology, University of Campania-Luigi Vanvitelli, Via Vivaldi 43, I-81100 Caserta, Italy. [2]Department of Bioorganic Chemistry, Leibniz Institute of Plant Biochemistry, Weinberg 3, D-06120 Halle, Germany. [3]Institute of Physical Biology, Heinrich Heine University (HHU) Düsseldorf, Universitätsstraße 1, D-40225 Düsseldorf, Germany. [4]Department of Physics, University of Pavia, Via Bassi 6, I-27100 Pavia, Italy. [5]Elettra Sincrotrone Trieste, Strada Statale 14 km 163.5, Area Science Park, I-34149 Trieste, Italy. [6]Department of NMR-based Structural Biology, Max Planck Institute for Multidisciplinary Sciences, Am Faßberg 11, D-37077 Göttingen, Germany. [7]Institute of Biological Information Processing, IBI-7: Structural Biochemistry, Forschungszentrum Jülich; Wilhelm-Johnen-Straße, D-52428 Jülich, Germany. [8]Department of Physics, University of Trento, Via Sommarive 14, I-38123 Povo Trento, Italy. [9]Department of Molecular Medicine, University of Pavia; Via Ferrata 9, I-27100 Pavia, Italy. [10]These authors contributed equally: Luigi Russo, Dipendu Dhar. ✉e-mail: Mehdi.Davari@ipb-halle.de; Nasrollah.RezaieGhaleh@unipv.it

potentially involved in XH·π (including NH·π) hydrogen bonds[13,14]. Nevertheless, the experimental support for their presence has been limited to a few CH·π interactions in aromatic and amide π systems of two folded proteins, where the small through-hydrogen bond J couplings between the C and H atoms of methyl groups and aromatic or carbonyl carbons could be detected[15–17]. Moreover, the presence of XH·π interactions on the solvent-exposed surface of proteins, where competing alternative modes of interactions are possible, remains elusive. In particular, their presence and functional role in IDPs are largely unknown, partly because of the lack of structural information.

Here, we report the direct experimental evidence of a π hydrogen bond present in an intrinsically disordered peptide, namely the Alzheimer's disease-related amyloid-β (Aβ) peptide. The reported π hydrogen bond exists between the peptide NH group of Gly22 and the aromatic π system of the preceding Phe20 residue in the Arctic (E22G) variant of Aβ (Fig. 1a). This NH·π hydrogen bond is detected through [1]H and [19]F NMR and Raman spectroscopic data and confirmed through observation of a hydrogen bond-mediated J coupling between the involved atoms, as predicted by the combined molecular dynamics (MD) simulations and density functional theory (DFT) calculations of the E22G-Aβ40 peptide.

## Results

### Upfield NMR chemical shifts of Gly22

Amide proton ($H^N$) and nitrogen (N) chemical shifts are sensitive probes of the local electronic environment of backbone NH groups of proteins[18,19]. Previous NMR chemical shift-based studies have shown that the wild-type (WT) Aβ40 is intrinsically disordered in the monomeric state and the structural effect of the E22G mutation is rather small and largely restricted to the vicinity of the mutation site[20,21]. Accordingly, the [15]N,[1]H correlation peaks of the six glycine residues of WT Aβ40 remain nearly unchanged upon E22G mutation, except for residue Gly25 close to the mutation site, which showed a small displacement (Supplementary Fig. 1). Strikingly, however, the correlation peak of the mutated residue Gly22 showed anomalous upfield chemical shifts (Fig. 1b and Supplementary Fig. 1a), which, as suggested previously[20,21], are probably caused by the ring current effect of

adjacent phenylalanine residues (Phe20 or Phe19). The ring shift of about −0.8 ppm for $H^N$ (and about −2 ppm for N) suggests the presence of an aromatic-amide interaction, in which the NH group lies roughly above the center of the aromatic ring[22].

### Near-zero temperature coefficient of Gly22's chemical shifts

Next, we measured the temperature coefficient of $H^N$ resonances in the WT and E22G-Aβ40 peptides. The temperature coefficients report whether the amide protons are involved in stable interactions, or alternatively engage in transient hydrogen bond interactions with surrounding water molecules and consequently their chemical shifts are influenced by temperature-induced changes in the hydrogen-bonded network structure of water. Overall, the $H^N$ temperature coefficients of WT Aβ40 were negative and large (Fig. 1b, c), indicating their predominant hydrogen bond interactions with water molecules[19,23,24]. The values and sequence profiles of the temperature coefficients were highly similar between the WT and E22G-Aβ40, except in the segment Phe20-Val24 around the mutation site (Fig. 1c), and in the segment His13-His14-Gln15-Lys16, most likely due to slight pH differences between the two samples. The most pronounced difference was observed for residue 22, where Gly22 of the E22G-Aβ40 peptide showed a near-zero temperature coefficient whereas Glu22 of WT peptide had a large negative value (Fig. 1b, c and Supplementary Fig. 1b). The combination of $H^N$ (and N) upfield chemical shifts and the near-zero temperature coefficient of Gly22 points to the existence of persistent interactions between the NH of Gly22 (residue i) and the π-electron cloud of a nearby aromatic moiety, most likely that of Phe20 at i-2 position (Fig. 1d), as suggested by previous surveys of protein structure databases[13]. Notably, in A21G-Aβ40, where Gly21 is preceded by Phe19, neither an upfield $H^N$ chemical shift nor a distinctive temperature coefficient was observed for Gly21 (Supplementary Fig. 2), indicating that the FxG sequence motif is not sufficient for formation of such NH·π interactions.

### Engagement of Phe20's ring in an interaction with Gly22

Next, we exploited the remarkable sensitivity of [19]F chemical shifts to their chemical environment[25,26] to probe the effect of the E22G mutation

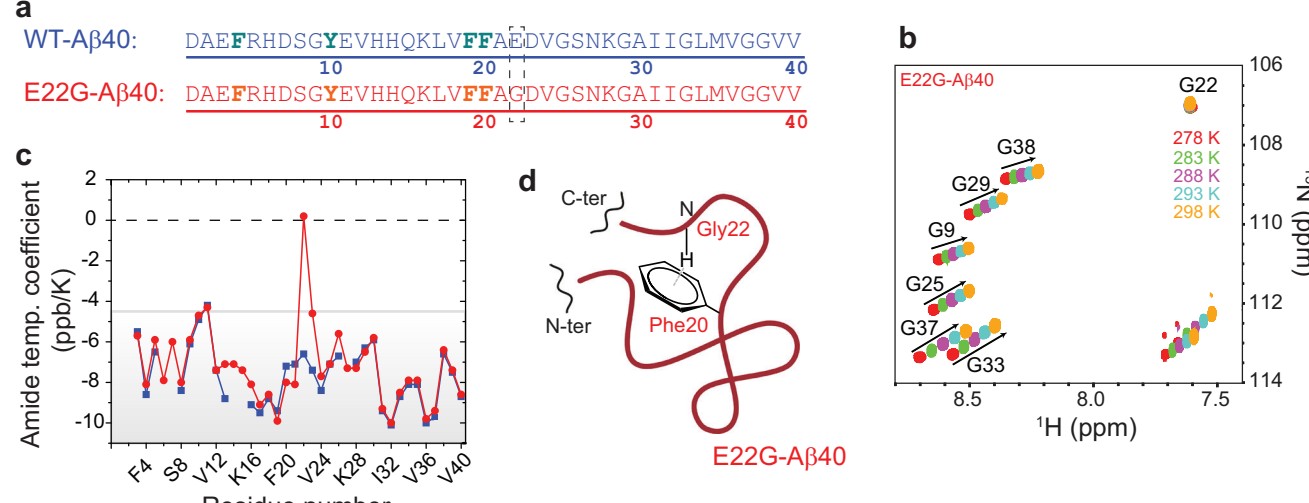

**Fig. 1 | NMR evidence for the engagement of residue Gly22 of E22G-Aβ40 in an NH·π interaction. a** Amino acid sequence of wild-type (WT) and E22G-Aβ40 in single letter code, with the mutation site and fluorinated amino acids highlighted by a dashed box and bold fonts, respectively. **b** [15]N,[1]H HSQC spectra of E22G-Aβ40 measured at temperatures 278–298 K, showing the remarkable lack of temperature sensitivity for the N and $H^N$ chemical shifts of Gly22, while the other six glycines exhibited large temperature sensitivity typical of unstructured proteins. **c** Residue-

specific temperature coefficients of $H^N$ chemical shifts for the WT and E22G-Aβ40. Residue Gly22 shows a near-zero temperature coefficient in the E22G-Aβ40 peptide. **d** Schematic representation of the suggested hydrogen bond between the amide proton of Gly22 and the aromatic side chain of Phe20 lying roughly beneath it (the cartoon of the peptide backbone created in BioRender. Russo, L. (2025) https://BioRender.com/jct0ak3).

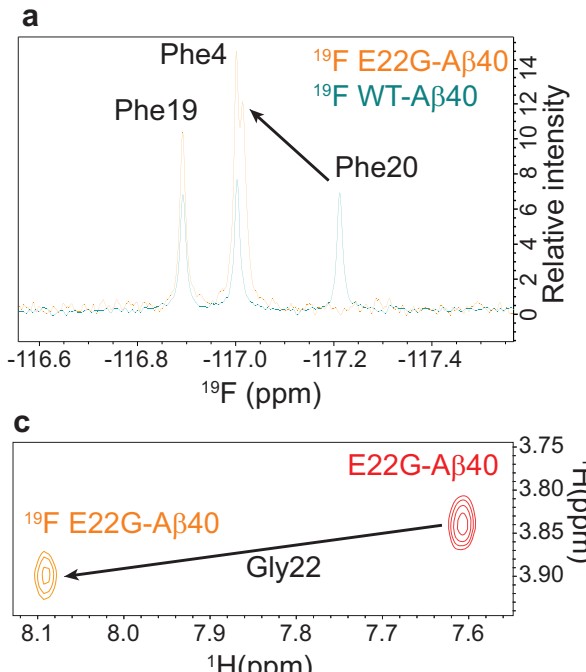

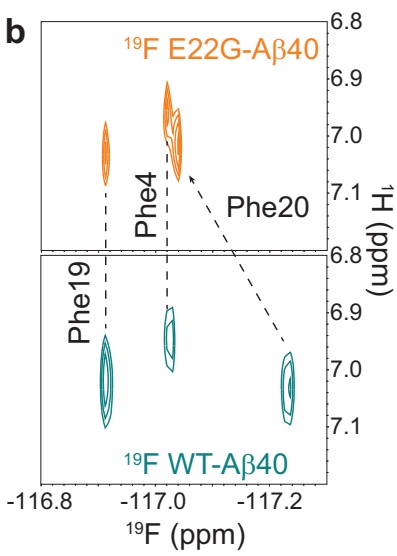

**Fig. 2 | NMR evidence for the engagement of residue Phe20 of E22G-Aβ40 in an NH·π interaction. a, b** Overlay of 1D ¹⁹F and 2D ¹H,¹⁹F HOESY spectra of ¹⁹F-labeled wild-type (WT, dark cyan) and E22G-Aβ40 (orange) peptides, showing ¹⁹F chemical shift perturbation of Phe20's signals. **c** Superposition of ¹H,¹H TOCSY spectra of ¹⁹F labeled (orange) and unlabeled E22G-Aβ40 (red), showing the downfield HN chemical shift of Gly22 in the ¹⁹F-labeled E22G peptide compared to the unlabeled peptide (8.09 vs 7.61 ppm). The loss of upfield chemical shift is caused by a reduction in diatropic ring current due to fluorine substitution.

on Aβ40's local conformation. To this end, we produced recombinant WT and E22G-Aβ40 with their three phenylalanine (Phe4, Phe19 and Phe20) and one tyrosine (Tyr10) residues labeled with fluorine atoms attached to the aromatic rings (Supplementary Fig. 3) and assigned ¹⁹F NMR signals through chemical synthesis of site-specific fluorine-labeled WT Aβ40 (Supplementary Fig. 4).The analysis of Hα chemical shifts showed that the backbone conformations of WT and E22G-Aβ40 were not altered by fluorine labeling (Supplementary Fig. 5). Interestingly, of the four ¹⁹F peaks observed in 1D ¹⁹F (Fig. 2a) and 2D ¹H,¹⁹F HOESY spectra (Fig. 2b), only the peaks belonging to Phe20's ring were significantly shifted by the E22G mutation. Furthermore, the ¹⁹F $T_1$ of Phe20 showed a small, but marginally significant, increase from 0.68 ± 0.02 to 0.73 ± 0.02 s, indicating its slight rigidification at nanosecond timescale upon the mutation (Supplementary Fig. 6, see also Supplementary Note in Supplementary Information).

The substitution of a hydrogen atom with the electronegative fluorine atom introduces a net partial positive charge at the *para* carbon atom, imposing a barrier against the diatropic ring current and reducing its magnetic shielding effect[27,28]. On the other hand, it is established that fluorine substitution slightly increases the electron density in the aromatic ring (due to resonance effect)[28] and could strengthen its potential role as an acceptor in XH·π hydrogen bonds. In line with the reduced magnetic shielding effect of fluorine-substituted rings[27], the H$^N$ chemical shift of Gly22 showed a much smaller upfield chemical shift in the fluorine-labeled E22G peptide than the unlabeled peptide (Fig. 2c). Overall, these findings indicate the spatial proximity of Gly22 and Phe20's ring in the E22G-Aβ40 peptide and the role of aromatic ring current effect in the upfield H$^N$ chemical shift of Gly22. A weak NOE correlation peak between the Hε of Phe20's ring and the HN of Gly22 further supported the spatial proximity of these two residues in the unlabeled E22G-Aβ40 (Supplementary Fig. 7).

**Altered vibration of Phe's ring in short mutated peptides**
To explore whether this NH·π interaction has influenced the vibrational dynamics of phenylalanine rings, Raman spectra were

collected for truncated tri- or tetra-peptide WT (FAE or FFAE) versus E22G (FAG or FFAG) Aβ (Supplementary Fig. 8). The Raman spectra of the WT peptides showed bands at 1006/1605 cm⁻¹ (in FFAE) or 1006/1607 cm⁻¹ (in FAE), which are respectively attributed to symmetric ring stretch $v_{12}$ and in-plane ring stretching $v_{8a}$ vibrational modes of the aromatic ring of phenylalanine[29]. Upon Glu to Gly substitution, the $v_{12}$ band showed a small red shift by 1 cm⁻¹ and the $v_{8a}$ band exhibited a clearer blue shift by 3 cm⁻¹ (Supplementary Fig. 8, see also Supplementary Fig. 9 for the reproducibility of the observed wavenumbers and shifts). The same trend of blue shift was observed in the DFT-calculated Raman spectra for the $v_{8a}$ vibrational mode (Supplementary Fig. 10). These small variations in the frequency position of the two Raman modes of $v_{12}$ and particularly $v_{8a}$ are consistent with the establishment of an interaction involving the aromatic ring of Phe, if present in the context of these short peptides. This can be expected considering that the frequency of the vibrational mode $v_{8a}$ is particularly sensitive to ring substituents[30].

**Characterization of the NH·π interaction via MD and DFT**
To further investigate the interaction between Gly22 and adjacent aromatic residues, MD simulations of E22G (and WT) Aβ40 (each 3 μs long) were performed using a99SB-disp force field[31]. This force field has been developed to model both ordered and disordered proteins across diverse systems and substantially improved accuracy in simulating IDPs without compromising performance on folded proteins. The MD ensemble thus generated under the infinite dilution assumption represents a reasonable approximation of the thermodynamic behavior of Aβ monomers under our experimental conditions, in which Aβ was kinetically stable against aggregation. Analysis of the MD trajectory using two distinct geometric cutoff values (distance $d_{CN}$ and angle θ: (4 Å, 45°) and (5 Å, 60°), see Supplementary Fig. 11 for details) for hydrogen bond interactions between amino acid residue 22 and π-acceptors revealed that the most significant and persistent NH·π interaction occurred between Gly22 and Phe20 (Fig. 3a and Supplementary Fig. 12).

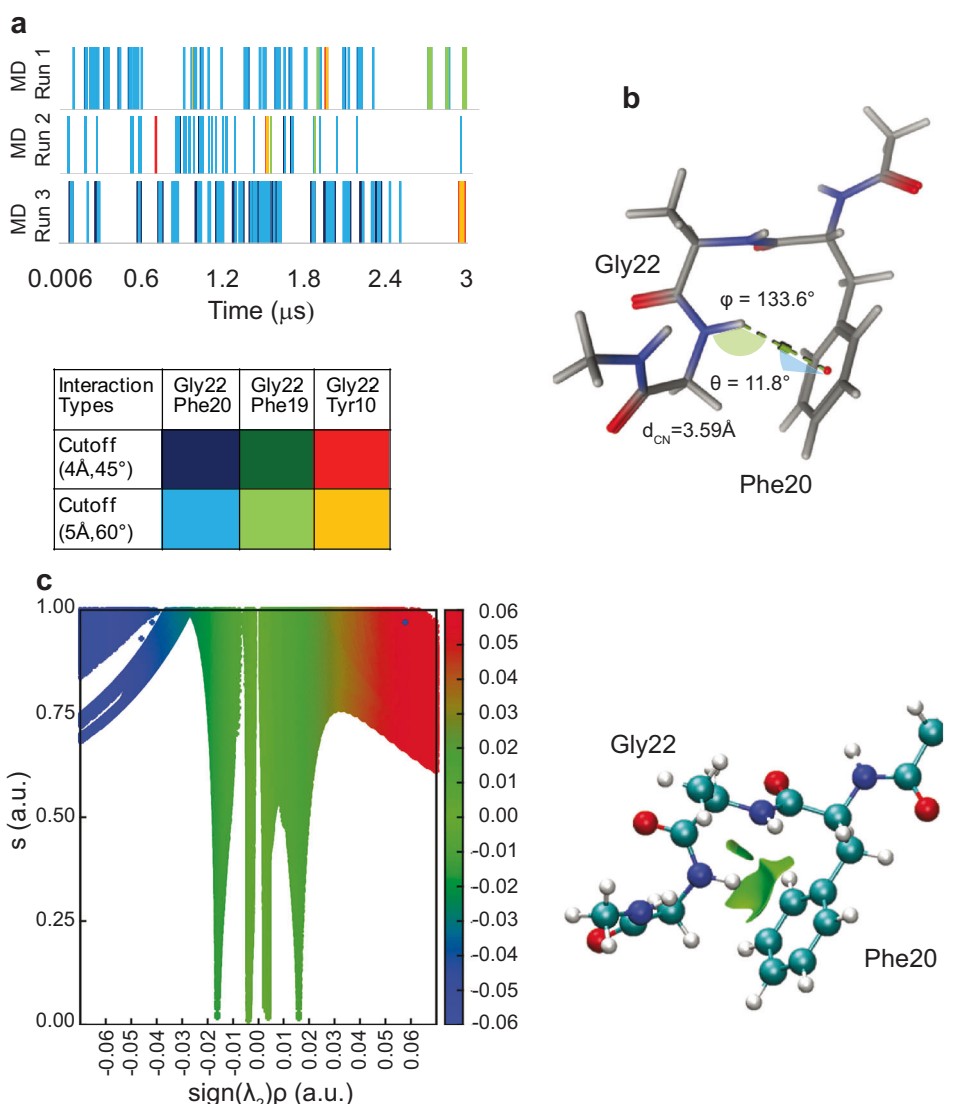

**Fig. 3 | Computational evidence for the engagement of Gly22 and Phe20 of the E22G-Aβ40 in an NH·π interaction. a** Molecular Dynamics (MD) simulation frames showcasing NH·π interactions observed in three independent MD runs of the E22G peptide (see Supplementary Fig. 12 for the WT peptide). The color scheme represents interaction types and their respective geometric cutoff values for distance $d_{CN}$ and angle θ. **b** The model chosen for density functional theory (DFT)-based calculations satisfies the stringent cutoffs of $d_{CN}$, θ and φ ($d_{CN} < 4.3$ Å, $-25° < θ < 25°$, and φ > 120°). **c** Non-Covalent Interactions (NCI) analysis of the computed electron density distribution (ρ) for the truncated FAG peptide. Left, the reduced density gradient (RDG, *s*) versus sign($\lambda_2$)ρ (see the text for definition; color scheme: blue, strong attractive interactions; green, weak interactions; red, strong repulsive interactions). Right, the RDG isosurface at *s* = 0.3 a.u.

To better understand and characterize the local non-covalent nature of NH·π interactions, we used a computational workflow including MD simulation and subsequent DFT calculations (see Methods for details). We selected the MD frames of the E22G peptide showing the most persistent NH·π interaction, extracted its Phe20-Ala21-Gly22 segment, and capped it at its N- and C-termini (optimized coordinates in Supplementary Table 1). Superposition of structures extracted from three independent replicates of MD simulations showed similar structures with a low RMSD of 0.406 Å (Supplementary Fig. 13), indicating that they lie within a narrow free energy basin. The selected frames fulfilled all three geometric criteria defined for an XH·π interaction (Fig. 3b)[13,32]. DFT calculations were then performed for these truncated tripeptide models using the hybrid B3LYP functional[33] with 6-31 + G(d,p) basis set[34] and empirical dispersion correction (GD3)[35]. The DFT-based wavefunctions were subsequently utilized for NCI analysis. The NCI analysis allows the identification and characterization of interactions on the basis of reduced (electron) density

gradient (RDG, *s*) isosurfaces[36]. As shown in Fig. 3c (and Supplementary Fig. 14), the light green spikes seen between $-0.01 < \text{sign}(\lambda_2)\rho < 0.01$ (where ρ is electron density and $\lambda_2$ is the second eigenvalue of the Hessian of $\rho_{(r)}$ with respect to position, **r**) indicate the presence of a weak interaction as a recurring feature in these structures. Similar results were observed for the fluorinated tripeptide model, suggesting that fluorine substitution at *para* position does not disrupt this interaction (Supplementary Fig. 15).

### Direct detection of the NH·π interaction via *J* coupling

Previous studies on classical and CH·π hydrogen bonds have demonstrated the presence of through-hydrogen bond *J* couplings between donor and acceptor atoms and their detection through NMR methods[7–9,15]. The DFT calculations performed at six different theory levels for the selected tripeptide model yielded total *J* coupling values of about 0.10-0.26 Hz between the amide H of Gly22 and the aromatic carbons of Phe20 (Supplementary Tables 2 and 3). The computed *J*

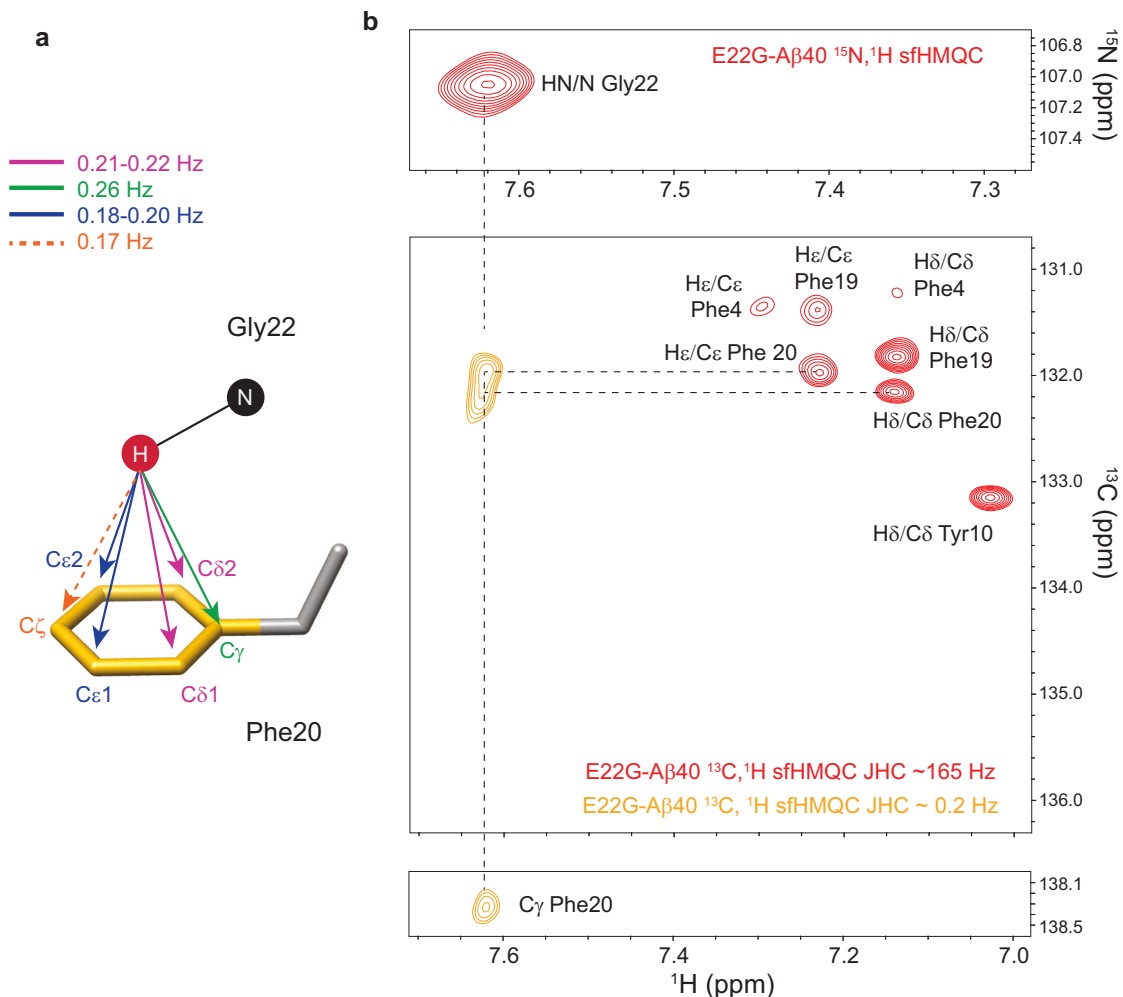

**Fig. 4 | Direct detection of through-hydrogen-bond scalar (*J*) coupling between Gly22 and Phe20. a** Predicted *J* couplings (-0.1-0.2 Hz) between the amide H of Gly22 and aromatic carbons of Phe20, obtained through density functional theory (DFT) calculations (Supplementary Table 2). **b** *J* coupling-based cross-peaks (with S:N of ~5-7) were observed between the amide H of Gly22 and Cγ, Cδ and Cε of Phe20 in the ¹³C,¹H SOFAST HMQC (sfHMQC) spectrum with very long transfer delays. The tentative correlation peak with Cζ was weaker (S:N of ~ 2, shown in Supplementary Fig. 18), possibly due to its smaller *J* value. The correlation spectra for directly connected nuclei, obtained through normal ¹³C,¹H and ¹⁵N,¹H sfHMQC spectra, are shown for the comparison of Cδ and Cε chemical shifts. The Cγ chemical shift matches the shift reported in ref. 38.

---

values are dominated by the diamagnetic spin-orbital (DSO) terms, which are partially canceled by the paramagnetic spin-orbital (PSO) terms. The contribution of Fermi contact (FC) and spin-dipolar (SD) mechanisms are relatively small. The relative contributions of these mechanisms are in contrast with the often dominant role of the FC mechanism in *J* couplings, including those mediated by the $CH_3$-π interactions as predicted in ref. 15, and could in principle reflect the different electronic configurations of the amide NH and the $CH_3$ groups. It is, however, notable that the FC contribution is very sensitive to the choice of the basis set and density functional[37], therefore, its actual contribution to the *J* couplings could be different from the values predicted here.

In principle, the predicted non-zero *J* values, however small they are, permit the transfer of magnetization between the involved nuclei in high-resolution NMR experiments. Nevertheless, the NMR-based detection of such small *J* values in proteins is challenging, because the NMR signals of proteins hardly survive the huge relaxation losses imposed by the long duration of transfer delays required for the evolution of such small *J* values. This problem becomes more serious for proteins such as Aβ, where the solubility and stability concerns often restrict their NMR sample concentration to <100 μM. To detect the predicted through-hydrogen bond *J* coupling between the amide

proton (of Gly22) and aromatic carbons (of Phe20), we performed carefully designed ¹H,¹³C SOFAST-HMQC (heteronuclear multiple quantum coherence) experiments (see Methods for details). After one week of measurement, no cross peak was observed in the obtained ¹³C,¹H spectrum of WT Aβ40 between Glu22's amide proton and Phe20's aromatic carbons (Supplementary Fig. 16). Excitingly, however, the spectrum of the E22G-Aβ40 revealed clear cross-peaks with S:N of ~ 5-7 correlating the amide proton of Gly22 and the Cγ, Cδ, and Cε of Phe20's ring (Fig. 4 and Supplementary Fig. 17)[38]. Despite their rather small S:N ratios, these cross-peaks were simultaneously observed at the expected chemical shifts of the involved nuclei, indicating that they are genuine signals. Besides, these cross-peaks were observed in three repeated measurements of freshly prepared E22G-Aβ40 samples, further confirming their genuineness. The tentative cross-peak with Cζ was weaker (S:N ~ 2, Supplementary Fig. 18), possibly due to the smaller size of its predicted *J* compared with other aromatic carbon atoms (Supplementary Table 2). Notably, the refocusing of *J* coupling through the introduction of two additional 180° proton-shaped pulses into the long-range SOFAST-HMQC pulse sequence led to the complete loss of these cross-peaks (Supplementary Figs. 19 and 20), substantiating the role of *J* coupling as the magnetization transfer mechanism underlying these correlations.

Possible alternative mechanism of residual dipolar coupling due to molecular magnetic susceptibility anisotropy could be excluded due to its negligibly small size (one to two orders of magnitude smaller than the predicted $J$ of ~ 0.1-0.2 Hz)[39,40]. The direct detection of $J$ coupling between the amide proton of Gly22 and several aromatic carbons of Phe20 further confirms the presence of an NH·π interaction between them.

### Prevalence of FAG and similar sequence motifs in IDPs

Next, we conducted a motif-based bioinformatics search across sequences of IDPs and intrinsically disordered regions (IDRs) from publicly available biological databases. The bioinformatics analysis of 7663 unique IDPs/IDRs showed that about 7% of the collected IDPs/IDRs exhibited the specific motif (Aro-Ala-Gly) or (Gly-Ala-Aro) identified in Aβ40 (Aro: Phe, Tyr, Trp; details of used databases in Supplementary Table 4). While the disordered nature of IDPs does not allow a structure-based search for these interactions, the rather high prevalence of this sequence motif promises the high likelihood of the occurrence of NH·π hydrogen bonds in IDPs.

## Discussion

Experimental and theoretical studies of XH·π interactions in small molecules have determined the geometric criteria underlying the formation of such energetically favored interactions[11]. In the case of NH·π interactions, the position of both N and H atoms above the aromatic ring seems to be important, with the interaction being strongest when the NH·π interaction is monodentate, that is, the N atom lies above the center of ring ($\theta = 0$) and the NH bond directly points at the center of the ring ($\phi = 0$), where the negative charge density is the highest[41]. Based on such geometric criteria, surveys of the protein structure databank (PDB) have suggested that XH·π interactions frequently exist in the tightly packed cores and binding interfaces of folded proteins[13,14]. Due to experimental challenges, however, the reports of verified XH·π interactions in proteins have been very rare. An interesting example was an NMR-based verification of CH·π interactions in ubiquitin and GB3 proteins, where very small DFT-predicted $J$-couplings in the range 0.07-0.31 Hz were detected between the C or H atoms of methyl groups and the carbons of Phe or Tyr aromatic rings[15]. In another report, the same group demonstrated CH·π interactions between methyl and carbonyl groups of ubiquitin[16]. There is no direct experimental report of NH·π interactions in proteins, and the only reported case was based on mutant cycle experiments in GB3 and an indirect calculation of interaction energies[17]. In this study, we report as a proof-of-principle one case of NH·π interactions found on the surface of an intrinsically disordered peptide, namely the E22G Aβ peptide, through direct detection of a scalar ($J$) coupling between the amide proton and several aromatic carbons (Fig. 4). While $J$ coupling could in principle be mediated through various non-covalent interactions (e.g., through weak van der Waals interactions[42]), the DFT calculated electron density, the predicted $J$ couplings, and the observation of $J$-mediated correlations between the amide proton of Gly22 and several carbons of Phe20's aromatic π ring support the underlying role of the NH·π interaction in mediating the observed magnetization transfer. Unlike XH·π interactions identified in the compact hydrophobic core of folded proteins, the detected NH·π interaction lies on the surface of the intrinsically disordered Aβ, where multiple alternative interaction modes with adjacent residues and water molecules are accessible. This situation resembles the XH·π interactions existing in small molecules dissolved in aqueous media, for which the interaction energy is predicted to be 1-4 kcal mol$^{-1}$, i.e., ca. 2-8 times the thermal energy at 298 K[11,43]. Although our DFT calculations suggest a weak interaction energy, likely due to a suboptimal tridentate configuration, this interaction is strong enough to keep the Gly22's NH group above the aromatic ring of Phe, thus preserving its upfield chemical shifts over a broad range of temperatures.

The AD-related E22G mutation increases the conformational heterogeneity of the formed Aβ fibrils[44,45]. Recent solid-state NMR and cryo-EM-based structures of the in vitro and brain-derived E22G Aβ fibrils do not show the NH·π interaction between Gly22 and Phe20, as detected in the monomeric peptide here[46,47]. Instead, they demonstrate a distinct stabilizing interaction between Phe20 and Val24 sidechains. We therefore hypothesize that the disruption of the Gly22-Phe20 NH·π interaction could introduce a kinetic barrier along the Aβ misfolding/fibrillation process and interfere with the conformational rearrangements in this strategic region essential for the ultimate formation of monomorphic ordered Aβ fibrils. In accord with this hypothesis, we propose that the altered ring dynamics of Phe19 and Phe20 residues could influence the highly plastic processes of E22G-Aβ misfolding and aggregation. Notably, the NMR relaxation rates of fluorine nuclei substituted at *para* positions of Phe rings are not sensitive to the aromatic ring flips[48], therefore, the effect of E22G mutation on Phe ring dynamics could be bigger than what is detected here.

Glycine and aromatic residues are relatively abundant in phase-separating IDPs[49], where the balance between intra- and multivalent intermolecular interactions can modulate the thermodynamics and kinetics of phase separation[50]. Furthermore, the conformational dynamics of glycine residues could influence the biophysical and material properties of the phase-separated condensates[49,51]. Our bioinformatics analysis shows that a significant fraction of IDPs contains glycine-based sequence motifs with a propensity to form intramolecular NH·π interactions, potentially acting as a mechanism for the regulation of phase separation. The combined experimental and computational approach presented here promises detection and characterization of such rather overlooked interactions in IDPs, even in systems as extensively studied as Aβ peptide. The reported XH·π interaction serves as a proof-of-principle example in IDPs, while direct experimental verification of NH·π interactions in folded proteins remains for future works. Further studies are warranted to clarify the general prevalence and potential impacts of XH·π interactions on the structure, dynamics, and function of IDPs. Moreover, the combined experimental and computational investigation of NH·π interactions in proteins with single well-ordered structures can provide additional key insights into the nature of these interactions beyond the current structure-based surveys.

## Methods

### Materials

$^{19}$F-labeled amino acids (4-fluoro-phenylalanine and 3-fluoro-tyrosine) with or without Fmoc groups were purchased from AnaSpec (California, US) and abcr, respectively. The synthetic unlabeled full-length Aβ40 peptides (wild-type, E22G, and A21G) and short peptide fragments (FFAE, FAE, FFAG, FAG, FFGE, FGE) were from Peptide Special Laboratory (PSL, Heidelberg, Germany) and used without further purification. The recombinant $^{15}$N,$^{13}$C labeled wild-type (WT) and E22G-Aβ40 were prepared as described before[21].

### Recombinant production of fluorine-labeled WT and E22G-Aβ40

Fluorine-labeled Aβ(1-40) was obtained by recombinant bacterial expression in defined medium based on the protocol of Gee et al.[52] using the hexa-histidine-tagged [NANP]$_{19}$ fusion construct described by Finder and Glockshuber[53]. Into this construct, the E22G mutation was introduced by mutagenesis PCR using primers with a gga (glycine) codon instead of gaa (glutamic acid) and Phusion® High-Fidelity DNA polymerase (New England Biolabs #M0530S). The PCR was prepared in triplicate, each having 50 µL total volume, 7.5 pmol forward Primer, 7.5 pmol reverse Primer, and 27 fmol methylated template plasmid. After 21 cycles (0.5 min denaturation at 95 °C, 0.5 min annealing at 55 °C, and 3 min polymerization at 72 °C), the products were treated with DpnI endonuclease (New England Biolabs #R0176S) for the removal of

leftover methylated template and cleaned up (PCR Cleanup Kit, Macherey-Nagel #740609.50). Due to overlapping forward and reverse primers, chemocompetent *E. coli* XL1 blue cells could be successfully transformed with the PCR product without enzymatic ligation prior to transformation. After monoclonal selection on LB agar with 100 mg/L ampicillin, the plasmid was amplified in the bacteria, extracted (Macherey-Nagel #740588.50), and sequenced. Finally, *E. coli* DL39(DE3) cells (Goldbio#CC-104-5×50) were transformed with the WT and E22G constructs and used for IPTG-induced expression in defined medium supplemented with 4-fluoro-phenylalanine (abcr #AB 101305) and 3-fluoro-tyrosine (abcr #AB 177447). The [NANP]$_{19}$-Aβ40 fusion construct was purified by IMAC in the presence of 6 M guanidinium and RP-HPLC. It was then lyophilized, resolubilized, and incubated with recombinant TEV protease to cut off the [NANP]$_{19}$ fusion partner. The Aβ40 was again purified by RP-HPLC after cleavage, and the final product was analyzed by RP-HPLC, SDS-PAGE, and Western Blot (in case of WT) and UHPLC-coupled ESI tandem mass spectrometry (performed at the Molecular Proteomics Laboratory, Medical Faculty and University Hospital Duesseldorf, Heinrich Heine University Duesseldorf). SDS-PAGE and analytical RP-HPLC detected no fragments or significant contaminations (Supplementary Fig. 3c). Western Blot and tandem MS confirmed the sequence identities, with the most abundantly detected species being in both wild-type and E22G peptides the desired full-length Aβ40, fluorinated at all four aromatic amino acid residues (Supplementary Fig. 3d, e). The recombinantly produced Aβ40 peptides were lyophilized, aliquoted in hexafluoroisopropanol, lyophilized again, and stored at −80 °C. For NMR experiments, the Aβ40 peptides were freshly solubilized in 20 mM (for the WT) or 50 mM (for the E22G mutant) NaOH and then brought to the final solution buffered with 20 mM sodium phosphate at pH 7.4 and used immediately.

### Chemical synthesis of fluorine-labeled WT Aβ40

Solid-phase chemical synthesis of fluorinated WT Aβ(1-40) was carried out in-house on the basis of Fmoc strategy on a H-Val-HMPB-ChemMatrix-resin (Sigma Aldrich), using an automated microwave synthesizer (Liberty 1, CEM). Before the chemical synthesis, the reactive side-chain hydroxyl group of Fmoc-3-fluoro-tyrosine was protected as *tert*-butyl ether by treatment with isobutylene and $H_2SO_4$ (performed at the Facility for Synthetic Chemistry, Max Planck Institute for Multidisciplinary Sciences, Goettingen, further details in an upcoming publication). Aβ(1-40) peptides with fluorine labels at four (Phe4, Phe19, Phe20 and Tyr10), three (Phe4, Phe19 and Phe20) or one (Phe4 or Phe19 or Phe20 or Tyr10) positions were synthesized and used for [19]F NMR resonance assignment.

### NMR experiments

NMR experiments were conducted at proton Larmor frequencies of approximately 600, 700, and 900 MHz using Bruker spectrometers equipped with cryogenic probes. Before NMR experiments, temperature calibration was performed using a standardized thermocouple or residual proton signal in a standard deuterated methanol sample. The temperature control to ±0.05 K during NMR experiments was achieved using the Bruker VT unit. Unless specified otherwise, the NMR samples contained 0.4 mg.mL$^{-1}$ of unlabeled, [19]F-labeled, or uniformly-[15]N,[13]C-labeled Aβ40 peptides (ca. 92 μM) dissolved in 20 mM sodium phosphate solution, pH 7.4. The NMR samples contained 10% (v/v) $D_2O$ for spectrometer frequency locking, and around 0.5 mM DSS for chemical shift referencing (0.000 ppm). Assignments for the [1]H and [15]N resonances of the WT and E22G-Aβ40 variants were taken from previous studies[21].

The chemical shift and NOE assignments of Aβ40 peptides were performed at 278 K by acquiring and analyzing the following standard spectra: i) 2D [1]H,[1]H TOCSY and NOESY of unlabeled peptides. The TOCSY experiments were performed using the DIPSI-2 spin-lock

sequence with mixing times of 60 and 75 ms. The 2D NOESY spectra were acquired with mixing times of 200 and 300 ms. ii) 2D [1]H,[15]N HSQC and 3D [1]H,[15]N NOESY-HSQC of [15]N,[13]C-labeled WT and E22G-Aβ40. A mixing time of 300 ms was used in the NOESY-HSQC experiment.

Hα secondary chemical shifts were obtained as the difference between the observed chemical shifts (δ$^{obs}$) and the residue-specific random coil values. The latter values were calculated using the approach defined by Poulsen and co-workers[54,55]. Temperature coefficients (Δδ/ΔT) for amide protons were calculated from the slope of the linear plot of H$^N$ chemical shift versus temperature. Amide proton chemical shifts for Aβ40 peptides were obtained from a series of [1]H,[15]N HSQC and [1]H,[1]H TOCSY spectra acquired at different temperatures ranging from 278 to 298 K.

[19]F NMR experiments were measured at a [19]F Larmor frequency of around 565 MHz (corresponding to proton Larmor frequency of around 600 MHz) and 283 K. An internal standard (TFA) was used for [19]F chemical shift referencing ( − 76.55 ppm). [19]F resonance assignment was made through comparison of 1D [19]F NMR spectra among site-specifically fluorine-labeled Aβ40 peptides. The [19]F $T_1$ relaxation times were measured through standard inversion-recovery experiments, with ten relaxation times exponentially distributed between 0.01 and 5.12 s and an interscan delay of 4 s. The [1]H, [19]F heteronuclear NOESY (HOESY) experiments were measured using a mixing time of 0.5 s and an interscan delay of 1.2 s. The [19]F $T_1$-based prediction of rotational correlation time (τ$_c$) was performed using the standard equations[26] and the previously reported reduced anisotropy values for fluorine-substituted aromatic rings.

The aromatic [13]C,[1]H SOFAST heteronuclear multiple quantum coherence (sfHMQC) measurements were performed at 600 MHz proton Larmor frequency and 278 K. Both WT and E22G-Aβ40 peptides were measured, with the WT measurement serving as a control experiment. The Aβ concentration was ~ 0.4 mg.mL$^{-1}$ (ca. 92 μM). The standard sfHMQC pulse sequence (Bruker) was slightly modified for the introduction of [15]N decoupling during direct [1]H acquisition. To detect directly connected aromatic [13]C and [1]H nuclei, a short transfer delay of 3 ms ( = 1/2 $J$, with $J$ of ~ 165 Hz) was used. To detect the weak hydrogen-bond mediated $J$ coupling between Gly22's H$^N$ and Phe20's aromatic carbons, a long transfer delay of 0.25 s was used, which assuming a $J$ of ~ 0.2 Hz (hence 1/2 $J$ of ~ 2.5 s), as predicted by DFT calculations (see Supplementary Table 2), allows ~ 10% evolution of the $J$ coupling and its consequent transfer of magnetization from Gly22's HN to Phe20's aromatic carbons. The choice of this delay was a trade-off between the evolution of $J$ coupling and the $T_2$ relaxation losses during the delay. This delay is still expected to be considerably bigger than the $T_2$ relaxation time of Gly22's H$^N$, hence leads to considerably more than 63% relaxation losses of the signal. To compensate for this huge relaxation loss, we used a very short recycle delay of 0.1 s (which is made possible thanks to accelerating $T_1$ relaxation of H$^N$ using a SOFAST experimental scheme) and 8192 scans per $t_1$ increment for 128 increments. Each long-range sfHMQC experiment took 7-8 days. The spectral width was 4225.81 Hz along $t_1$ and 5341.88 Hz along $t_2$.

The 2D and 3D NMR spectra were processed by NMRpipe[56] or Topspin 4.0.1 (https://www.bruker.com/) and analyzed using NMRFARM-SPARKY[57] and CARA[58] software.

### Raman spectroscopy measurements

For Raman measurements, aqueous solutions of tri- and tetra-peptide fragments of Aβ40 corresponding to WT (in single-letter code: FFAE, FAE) or E22G (FFAG, FAG) were prepared and deposited on a glass slide, waiting for a few minutes until most of the water molecules are evaporated. Raman spectra were collected in the wavenumber range between about 400 and 4000 cm$^{-1}$ using an integrated micro-Raman system (Horiba Jobin Yvon, LabRam Aramis). The excitation radiation at 532 nm was focused onto the sample surface with a spot size of about 1 μm$^2$ through a 100X objective. The Raman signal was collected

in back-scattering geometry, analyzed through a 46 cm focal length spectrograph equipped with a holographic 1800 grooves/mm grating, and detected with a charge-coupled device (CCD) detector. The dispersion was set at 0.8 cm$^{-1}$/pixel. The elastically scattered radiation was filtered by using a narrow-band edge filter. The Raman spectra were recorded on the same sample several times to ensure the reproducibility of the measurements and to exclude any possible photodegradation effect.

## Molecular dynamics (MD) simulations

The structure of WT and the E22G variant of the peptide Aβ40 was constructed as a linear model in MOE 2022.02. The protonation states of titratable amino acid residues were assigned according to PROPKA2[59] pKa calculations and visual inspection. The computational models of Aβ40 (both WT and E22G) were used as initial structures to perform atomistic Molecular Dynamics (MD) simulations using GROMACS software suite 2022.1[60]. MD simulations were run at infinite dilution to model the thermodynamic ensemble of monomeric Aβ40 (critical concentration for Aβ40 aggregation is 0.1–1 μM). To emulate infinite dilution, the peptide was thermodynamically isolated by maintaining a minimum distance of 2.45 nm between the peptide and the edges of the simulation box, while truncating long-range non-bonded interactions at 1.0 nm. The a99SB-disp force field having optimized solute–solvent dispersion interactions[31] and TIP4P water model[61] were used. This force field was developed by Shaw et al.[31] and includes optimized torsion parameters and modifications in the protein and water van der Waals interaction terms to the Amber99SB force field, which dramatically increases the accuracy of disordered protein simulation without sacrificing accuracy for folded proteins. The a99SB-disp force field is shown to reproduce both secondary-structure propensities and global conformational heterogeneity of disordered peptides.

Both the WT and the corresponding E22G variant of Aβ40 were initially converted from their respective pdb format into a standardized GROMACS format and centered in a rectangular solvation box of $15 \times 6 \times 6$ nm dimension with periodic boundary condition (PBC). The system was equilibrated at 300 K and 1 bar for 0.1 μs using a standard two-step protocol—first an NVT phase to stabilize temperature, followed by an NPT phase to equilibrate pressure and density until convergence was achieved. The force field parameters were applied, including bond constraints using LINCS algorithm[62] for maintaining bond lengths, and neighbor searching was performed with the Verlet integration scheme[63]. Long-range electrostatic interactions were treated using the Particle Mesh Ewald (PME) method[64] with a cubic interpolation order of 4. Long distance Van der Waals and electrostatic interactions were truncated at 1 nm. Temperature coupling was achieved using a modified Berendsen thermostat[65] with separate temperature groups for protein and non-protein molecules, at a reference temperature of 300 K and a time constant of 0.1 ps. Pressure coupling was achieved using the Parrinello-Rahman method[66] with an isotropic scaling of box vectors, at a reference pressure of 1 bar and a time constant of 2 ps. Production runs were performed at 300 K using a 2 fs time step for 3 μs. The production simulations were conducted independently in triplicate, each initiated with a randomly generated velocity. The complete system setup is provided in Supplementary Table 5. The MD trajectories were analyzed using MDAnalysis 2.9.0[67,68] Python package for NH-π interaction between Gly22/Glu22 and the aromatic residues in WT and E22G. The MD trajectories were visualized and analyzed in VMD version 1.8.6[69].

## Geometric cutoff criteria for NH-π hydrogen bond detection

To elucidate the geometric criteria governing NH-π hydrogen bond interaction, we followed the statistical analysis of geometric parameters using scatterplots similar to the ones previously used by Steiner et al. in their studies[13]. These plots in Supplementary Fig. 11 show

the angle θ and distance $d_{CN}$ between the NH group of Gly22 and the Phe20 residue (donor and acceptor of the hydrogen bond). The angle θ was defined between a reference vector passing perpendicularly through the centroid of the benzene ring of Phe20 and the vector connecting this centroid C to the donor N of NH (also used for distance $d_{CN}$ calculation), as shown in Supplementary Fig. 11a. The scatterplot of distribution for three independent MD runs was visualized in Supplementary Fig. 11b. The plot indicates that most interactions occur at linear angles to the reference vector or from similar opposite directions. Interactions between 60° and 120° were rare, likely corresponding to the aromatic ring's edges where the π cloud is minimal, limiting H-π interactions.

Adopting this angular criterion for θ, we plotted the distribution of probable occurrences of H-π interactions for distances $d_{CN}$ with a bin size of 0.1 Å, as shown in Supplementary Fig. 11c. This plot revealed a distinct maximum at around 3.5 Å, with the shortest contacts being approximately 3.1 Å and the median being approximately 3.8 Å. Previous studies[13,70,71] have suggested $d_{CN} < 4.5$ Å and $-25° < \theta < 25°$, as a reasonable cutoff for this interaction. Building upon this concept, we utilized two more flexible distance $d_{CN}$ and angle θ cutoff values of (4 Å, 45°) and (5 Å, 60°) for further analysis shown in the red dotted line in Supplementary Fig. 11c. The Gly22-Phe20 interaction of the E22G variant of Aβ40 was selected for calculating these criteria as it was assumed to be the most frequently occurring case, considering the experimental findings. The codes are available at https://github.com/davari-group/MD-Distance-Angle-Analysis.

It is notable that although our force field was not explicitly parameterized for NH−π recognition, our microsecond-scale MD trajectories capture these NH−π interactions under our stringent geometric criteria. Their transient nature of interaction is treated statistically via clustering and geometrical analysis; thus, the results reflect features of the ensemble free-energy landscape rather than isolated short-lived encounters. To characterize the short-range noncovalent interactions, representative MD snapshots meeting the desired geometric criteria were extracted and subjected to subsequent DFT geometry optimization and further analysis.

## Density functional theory (DFT) calculations

All DFT calculations were carried out using Gaussian 16 Rev. C.02[72]. Structural snapshots from the MD trajectory with observable Gly22-Phe20 interaction within the geometric cutoff of $d_{CN}$ and θ (4 Å, 45 °) for an NH-π hydrogen bond interaction were randomly extracted from all three independent MD runs. It is worth mentioning that the DFT calculations were applied to peptide models relevant to the monomeric equilibrium state. The truncated peptides containing residues 20, 21, and 22 were extracted from these frames using VMD as PDB structures. These structures were capped with acetyl (ACE) and N-methyl (NME) groups at the N- and C-termini, respectively, in MOE[61]. These truncated structures were inspected and visualized in GaussView[73] and the added cappings were frozen before DFT calculations. The geometry optimization and vibrational frequency calculations were performed with the hybrid B3LYP functional[33] and with 6-31 + G(d,p) basis set[34] having empirical dispersion correction (GD3) (see Supplementary Table 1 for the coordinates of optimized models)[35]. The self-consistent reaction field SCRF(IEFPCM)[33] was used as an implicit solvent model. It is notable that DFT was then applied, not to predict the global thermodynamic minimum, but to refine the local geometry and enthalpic strength of specific NH−π contacts within thermally populated snapshots.

Further, DFT-based calculations were performed for non-covalent interactions (NCI) characterization and computation of the Raman spectra and indirect nuclear spin−spin (J) couplings. The wavefunction of each molecule was produced by Gaussian at the same level of theory. The wavefunctions extracted from the Gaussian calculations were further used to perform NCI analysis of NH-π interactions using

NCIPlot[74]. To study the effect of fluorine substitution on the NH-π interaction, the above process was repeated after substituting *para* H with F atom in Phe20. The NCI results were visualized in VMD and a graphical representation of the reduced density gradient (RDG, *s*) vs sign($\lambda_2$)$\rho$ plots were generated using Gnuplot 4.4[75].

The Raman spectra were calculated for the peptides based on vibrational frequency calculation at the same level of theory. Raman spectra were calculated for peptide models with the explicit water molecules. The snapshots of the peptides showing the NH-π interactions at the geometric distance $d_{CN}$ and angle θ cutoff values (4 Å, 45°) with interacting water molecules in the first hydration shell of 4 Å were extracted from MD trajectories and used for calculating these Raman spectra. Visual assessment of the selected vibrational mode of the phenylalanine residue was carried out using GaussView 6[74]. These results were analyzed in Multiwfn program 3.8[76] and the corresponding Raman spectra data were extracted without application of any scaling factor and plotted using a script written in Python 3.12. The DFT-predicted and experimental Raman spectra of the truncated WT (FAE) and mutated (FAG) tripeptides were compared in order to assess the qualitative agreement between the DFT and experiment regarding mutation-induced Raman band shifts.

DFT calculations of indirect nuclear spin–spin (*J*) coupling constants and the individual contributions of Fermi contact (FC), spin-dipolar (SD) and paramagnetic and diamagnetic spin-orbit (PSO, DSO) mechanisms were carried out using the B3LYP functional and the 6-311++G(d,p) basis set on all atoms, as in previous reports[15]. To examine the effect of the chosen basis set and functional on the predicted *J* couplings, DFT calculations of *J* couplings were repeated at five additional levels of theory: B3LYP/6-31++G(d,p), PBE1PBE/6-311++G(d,p), PBE1PBE/cc-pVTZ, PBE1PBE/aug-cc-pVTZ, and wB97XD/aug-cc-pVTZ (Supplementary Tables 2 and 3).

### Prevalence analysis of NH-π interactions in IDPs/IDRs
All IDPs/IDRs present in the related protein were collected (see Supplementary Table 4 for details). The IDP protein sequences were extracted from these data and merged. All redundancies were removed from the merged dataset. We defined a threshold of 10 residues as the lower limit and kept the upper limit free to search for the sequence motif among 7663 unique IDPs/IDRs from the preprocessed dataset. A general motif was defined as aromatic residues (Aro: Phe, Tyr, Trp) followed by any residue, and then finally glycine, or its reverse (Aro-x-Gly or Gly-x-Aro). Pertaining to the importance of the middle residue in the motif on the NH-π interaction, a more rigid specific motif was defined as Aro-Ala-Gly or Gly-Ala-Aro. The peptide sequences with the general and the specific motif were identified. The frequency of repetition of the specific motif for each peptide was also calculated.

### Reporting summary
Further information on research design is available in the Nature Portfolio Reporting Summary linked to this article.

## Data availability
The molecular simulations generated in this study have been deposited in the Zenodo database under accession code 16737783[77]. DFT optimized coordinates for the peptide models are given in SI. The raw NMR data (plus NMR acquisition parameters) have been deposited in the Zenodo database under accession code 17410262[78]. Source data are provided as a Source Data file. Source data are provided with this paper.

## Code availability
The custom codes used in this study are deposited in the Zenodo database under accession code 16737783[77] or available in GitHub (https://github.com/davari-group/MD-Distance-Angle-Analysis or https://doi.org/10.5281/zenodo.17417064)[79].

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

## Acknowledgements

N.R.-G. acknowledges Deutsche Forschungsgemeinschaft (German Research Foundation, DFG) for research grants RE 3655/2-1 and RE 3655/2-3. M.D.D.'s research is supported by funding from IPB Halle. We acknowledge Elettra Sincrotrone Trieste for providing access to its synchrotron radiation facilities and for financial support under the IUS internal project (proposal number 20230190). We thank Prof. Pietro Galinetto for his support regarding Raman measurements, Dr. Lothar Gremer for technical suggestions in recombinant preparation of $^{19}$F-labeled Aβ peptides, Dr. Anja Stefanski from Molecular Proteomics Laboratory, Medical Faculty and University Hospital Duesseldorf, Heinrich Heine University Duesseldorf, for mass spectrometry experiments and analyses, and the Synthetic Chemistry facility of the Max Planck Institute for Multidisciplinary Sciences, Goettingen, for the synthesis of protected Fmoc-fluoro-Tyr used for chemical synthesis of fluorinated Aβ.

## Author contributions

Conceptualization: N.R.-G. Methodology: N.R.-G., M.D.D. Investigation: N.R.-G., L.R., D.D., M.D.D., B.R., O.P., F.M. Visualization: L.R., D.D., N.R.-G., R.B., B.R., M.D.D., F.M. Resources: N.R.-G., M.D.D., B.R., R.B., K.O., K.G., S.B., D.W., C.G., L.R. Funding acquisition and Project administration: N.R.-G., M.D.D. Supervision: N.R.-G. Writing – original draft: N.R.-G. Writing – review & editing: All authors.

## Funding

## Competing interests

The authors declare no competing interests.
