## [Transparent Peer review file · Nature Communications]

Direct detection of an NH- π hydrogen bond in an intrinsically disordered peptide

Corresponding Author: Dr Nasrollah Rezaei-Ghaleh

Version 0:

Reviewer comments:

Reviewer #1

(Remarks to the Author)

This is a very well written and meticulous paper with the central result being the observation of a cross-H bond NH- π J coupling in a peptide. This appears to be the first convincing measurement of such a coupling, implying an interaction between the fragments. The work builds on previous NMR work looking at other cross-hydrogen bond J couplings. The NMR work is the highlight of this paper and is of excellent quality. The result is of broad interest. I have the following comments for the authors to consider:

1. The authors state that the observation of J coupling implies a covalent character to the bond. This is unfortunately not true and should be removed. See for example <https://doi.org/10.1063/1.1332994> which clearly establishes that J couplings can be observed in the absence of any covalent character.
2. The authors rightly state that the DSO mechanism is the largest. It is often the case that the DSO and PSO mechanisms partly cancel each other, and that indeed the FC mechanism governs the final result to a large extent. In this case, the FC part just happens to be small. See for example Figure S2(a) of the authors' reference 15 which presents a similar case where the DSO and PSO mechanisms are larger than the FC, but the FC dominates the trend in J coupling. The discussion in the present manuscript could perhaps be adjusted to reflect this.
3. Reference <https://doi.org/10.1002/anie.201702626> appears to be relevant to the current work and could also provide some insights.

Reviewer #2

(Remarks to the Author)

Being a non-expert in proteomics, I see the following points the manuscript could potentially be improved.

The authors claim their experimental results supported by DFT and MD simulations demonstrated the existence of NH- π bonds.

To me, there are a couple of questions:

- (1) The authors appear to believe the observation of J coupling in NMR and Raman should be taken as conclusive evidence that such couplings are mediated by NH- π bonds. Without in-depth knowledge in NMR, I am not sure if this is a unique interpretation. It seems to me that J coupling could be mediated by other mechanisms. I believe it would be appropriate for the authors to make this point clearer.
- (2) The Raman peak-position based deductions rely on peak position shift by 1 to 3 cm^{-1} . While the spectra presented seem to have very low noise, I wonder if those are the raw data or are those after background subtraction and noise removal. If they make multiple Raman spectra harvesting, a figure including a superposition of say 10 spectra of each sample should be presented. Such a plot would give the readers a sense of the uncertainty in peak position determination.
- (3) DFT calculations render such a high degree of agreement with the experimental data warrants explanation. This is because experimental results are obtained at room temperature and in the presence of statistical fluctuations such as Brownian motion whereas DFT could only provide the thermal equilibrium configuration at zero degrees kelvin. A discussion about the expected differences is due with an emphasis on the anticipated "proximity to thermodynamic equilibrium configuration" of the peptides under the experimental condition, and thus the anticipated agreement between the DFT results and the experimental results. In other words, how "far" should we expect the configuration of those peptides be from

the true equilibrium? An easier to understand way of asking the question would be: If the peptides were left in the analyte for indefinitely long time, would their configuration (secondary, tertiary, etc) be the same as reported in this study? Would they coagulate or precipitate out? If so, would those be the configuration closer to thermal equilibrium than those studied, and shouldn't the DFT study predicts that configuration?

(4) MD simulation requires concrete knowledge of the interaction potentials and for that matter the overall potential energy landscape. That, in the case of individual IDP molecules subjecting to random fluctuations including finite temperature and Brownian motion, and others, should be expected to be a fluctuating factor during the simulation time duration of microseconds. In view of this consideration, the applicability of MD simulation to predicting the experimental observations also warrants further elaboration. In other words, the applicability of MD simulation in this study needs to be rigorously justified.

These are just some of the questions that come to my mind after reading the manuscript. I will not be able to provide a substantial opinion about whether the manuscript is suitable for publication in Nature Comm, due largely to my lack of in-depth knowledge pertaining to NMR and also to proteomics.

Reviewer #3

(Remarks to the Author)

This work provides a proof-of-principle for the existence of NH- π hydrogen bonds in proteins. The authors studied the NH- π hydrogen bond between the amide proton of Gly22 and the aromatic carbons of Phe20 in the E22G mutant of Abeta40. They used a combination of NMR spectroscopy, including ^{19}F labelling, Raman spectroscopy, MD simulations and DFT to provide evidence for the presence of this hydrogen bond.

The critical issue with this work is that a general claim is made based on a single example. It remains unclear whether NH- π hydrogen bonds generally exist in proteins, and if they do, how much they contribute to the conformational stability and other properties.

To detect the J coupling corresponding to the NH- π hydrogen bond, the authors use a SOFAST-HMQC with ultralong delay approach. This is ingenious and represents state-of-the-art practice for detecting sub-Hz heteronuclear J couplings in biomolecules. However, the measurements are at the technical limit of NMR sensitivity, making the signal-to-noise borderline to extract conclusions, given possible confounding factors, such as for example residual dipolar couplings or other magnetization transfer mechanisms.

These technical challenges seem to have guided the choice to use a disordered protein for their proof-of-principle study. However, the existence of conformational disorder introduces problems with the interpretation of the results. It would be important to establish the existence of NH- π hydrogen bonds also in folded proteins.

The use of the model tripeptide Phe20-Ala21-Gly22 in the DFT calculations is likely to overestimate the interaction energy. Since the study concerns a disordered protein, it is not appropriate to analyze a single configuration of this peptide. DFT results should be reported for multiple configurations.

Version 1:

Reviewer comments:

Reviewer #1

(Remarks to the Author)

The authors have taken all reviewers' comments seriously and addressed them thoroughly. Additional computational and experimental data have been added to solidify the claims made in the paper. Claims have been adjusted appropriately and are supported by the data. It is true that this still only represents one experimental example as pointed out by reviewer 3, but the authors have done a good job in addressing this point.

Overall the work is sound and conclusions are not overstated.

(Remarks on code availability)

Reviewer #2

(Remarks to the Author)

I have to say that the response of the authors left still quite a few questions unanswered. Below I list my remaining questions.

(1) The authors cited once again the reasons they think the observed J-coupling should be due to NH- π bonds. What would be helpful is if they provided arguments why other cause for J-coupling being unlikely. They seem to believe DFT calculation to be strong supporting evidence without a satisfactory answer to my question in (3) questioning the validity of the accuracy (see below for more details).

(2) Comparison of two consecutive measurements of FFAE, for example, leads to noticeable questions regarding the assurance provided by the authors. The signal-to-noise ratio of the red and black spectra are obviously different. However, in the context of how the authors used Raman spectra to justify the trustworthiness of their DFT results pertains to the Raman peak position shift of $1-3\text{ cm}^{-1}$, they need to provide say 10 superimposed Raman spectra to provide a quantitative

reproducibility of the peak positions including the wavenumber increment size. The information would help the readers to judge the certainty when comparing DFT with Raman measurements and thus the significance of the 1-3 cm⁻¹ peak position shifts and is in my opinion absolutely necessary.

(3) The authors seem to have misunderstood my question about whether one should expect DFT calculations to render that degree of agreement with the experimental results given they are supposedly accurate for samples under totally different temperatures (300 K vs 0 K). In other words, at the core of the question is the role played by entropy. The argument they provided talks about the kinetics instead of thermodynamics of the sample system. One case in point is that just because a piece of glass does not incur significant change over a time period of say 10 years does not in any way infer the proximity of the glass to thermodynamic equilibrium, i.e. the single crystalline state of SiO₂, which is the state relevant to DFT calculations.

(4) My final question was directed at the validity of the potential or the force field as the authors call it, used in their MD simulation, and the time variability of the force field. I must once again make it clear about my lack of specific knowledge background of the field of sciences in discussion. My question stems from the consideration of the molecular scale environment of IDP molecules in solution. Given that the molecule-molecule interaction becomes significant only when two molecules are in close proximity (~2.5 nm as pointed out by the authors) to one another, and that at this scale the environment and thus the potential or the force field is expected to fluctuate significantly with time, I believe the authors are obligated to provide convincing arguments why the force field they employed is valid not pertaining to the experiments in the cited references but to the formation and dissociation of NH- π bonds.

While I did not want to review the rebuttal, the reply of the authors gave me an uneasy feeling about the substance of this work and thus these follow up questions.

(Remarks on code availability)

See above

Reviewer #3

(Remarks to the Author)

The present work now convincingly documents one NH- π case in an IDP. While the general claims are now tempered, and the challenge and independent scope of studies on folded proteins are explicitly acknowledged, there remains no experimental evidence of NH- π bonds in folded proteins, where interpretation would be less confounded by conformational heterogeneity.

The Title, Abstract, and Conclusions should unambiguously state that this proof-of-principle is for an intrinsically disordered system and avoid implying general prevalence in folded proteins beyond citing structure-survey expectations. It may help to add one sentence to the Abstract noting that validation in folded proteins remains future work.

(Remarks on code availability)

Version 2:

Reviewer comments:

Reviewer #1

(Remarks to the Author)

The authors have taken the reviews seriously and addressed all comments. Substantive detail has been provided and appropriate changes have been made. I recommend the paper be published.

(Remarks on code availability)

We thank the three reviewers of our manuscript for their positive and encouraging remarks, insightful comments, useful suggestions and constructive critiques, which, in our view, have helped us to significantly strengthen our manuscript. Below, please find our point-to-point response to the comments made by each reviewer.

Reviewer #1 (Remarks to the Author):

This is a very well written and meticulous paper with the central result being the observation of a cross-H bond NH- π J coupling in a peptide. This appears to be the first convincing measurement of such a coupling, implying an interaction between the fragments. The work builds on previous NMR work looking at other cross-hydrogen bond J couplings. The NMR work is the highlight of this paper and is of excellent quality. The result is of broad interest.

Response: Many thanks for the positive evaluation and encouraging words by the reviewer.

I have the following comments for the authors to consider:

1. The authors state that the observation of J coupling implies a covalent character to the bond. This is unfortunately not true and should be removed. See for example <https://doi.org/10.1063/1.1332994> which clearly establishes that J couplings can be observed in the absence of any covalent character.

Response: We thank the reviewer for raising this point and drawing our attention to this interesting reference. Our argument for the partial covalent character of this NH- π interaction was grounded on the computed electron densities (as shown, e.g., in Fig. 3c) and detection of J coupling-based magnetization transfer between the amide proton (of Gly22) and several aromatic carbons (of Phe20), in line with its π character. We are aware that the strict quantum mechanical definition of “covalence” is still under debate and admit that the J coupling-based magnetization transfer *per se* does not prove the covalent character of the studied interaction. We thank again the reviewer for highlighting this crucial point. To address the reviewer’s comment, we have now deleted our previous sentence regarding the “partial covalent” character of the detected NH- π interaction from the Discussion section and removed the QTAIM data related to quantifying the covalence of this interaction from the Results part. We have also removed the sentence “The observation of a J coupling is the ultimate proof of a hydrogen bond” from Introduction and toned down our emphasis in Results, p. 11, by removing the phrase “ultimate proof”, as follows:

“The direct detection of J coupling between the amide proton of Gly22 and several aromatic carbons of Phe20 further confirms the presence of an NH- π interaction between them.”

The mentioned reference has also been added as ref. 42.

2. The authors rightly state that the DSO mechanism is the largest. It is often the case that the DSO and PSO mechanisms partly cancel each other, and that indeed the FC mechanism governs the final result to a large extent. In this case, the FC part just happens to be small. See for example Figure S2(a) of the authors' reference 15 which presents a similar case where the DSO and PSO mechanisms are larger than the FC, but the FC dominates the trend in J coupling. The discussion in the present manuscript could perhaps be adjusted to reflect this.

Response: As said by the reviewer, the predicted contribution of the FC mechanism to the $J_{\text{HN-C}}$ coupling in our study is small, and the total $J_{\text{HN-C}}$ coupling is dominated by the net contribution of DSO and PSO mechanisms. This is in contrast with the predicted $J_{\text{HC-C}}$ coupling reported in ref. 15, where the FC mechanism dominates the total J coupling value and its distance-dependent trend. While this discrepancy could in principle reflect the difference between the electronic configuration of the amide NH group in our work and the CH₃ group in theirs, it is also possible that this discrepancy arises due to the specific choice of density functional/basis sets in these studies. To examine this possibility, we have now performed DFT calculations of J couplings at six different levels of theory (density functional/basis sets of B3LYP/6-31++g**, B3LYP/6-311++g**, pbe1pbe/6-311++g**, pbe1pbe/cc-pVTZ, pbe1pbe/aug-cc-pVTZ or wB97XD/aug-cc-pVTZ), including the one used in ref. 15. The results are now included as Extended Data Tables 1 and 2 in the revised manuscript. Although the predicted J values exhibit some degree of variation in their exact values, the general trends are consistent: that the total $J_{\text{HN-C}}$ coupling values are in the range of 0.10-0.26 Hz, that the dominant part of total J is contributed by the net effect of DSO and PSO mechanisms, and that the FC and SD contributions are relatively small. We would however like to emphasize that these calculations were not intended to provide a quantitative picture of J couplings and their underlying mechanisms in the studied peptide. Instead, we were only interested in showing the plausibility of non-zero J couplings between the nuclei engaged in the NH- π interaction, then verifying its presence experimentally. To address the reviewer's comment, we have now included the new data in the revised manuscript (as Extended Data Tables 1 and 2) and modified the Results section, pp. 9-10, as follows:

“The relative contributions of these mechanisms are in contrast with the often dominant role of the FC mechanism in J couplings, including those mediated by the CH₃- π interactions as predicted in ¹⁵, and could in principle reflect the different electronic configurations of the amide NH group and the CH₃ groups. It is however notable that the FC contribution is very sensitive to the choice of the basis set and density functional ³⁷, therefore its actual contribution to the J couplings could be different from the values predicted here.”

3. Reference <https://doi.org/10.1002/anie.201702626> appears to be relevant to the current work and could also provide some insights.

Response: Thanks for introducing this reference. We have now cited it as ref. 16 in the Introduction, p. 4:

“Nevertheless, the experimental support for their presence has been limited to few CH- π interactions in aromatic and amide π systems of two folded proteins, where the small through-hydrogen bond J couplings between the C and H atoms of methyl groups and aromatic or carbonyl carbons could be detected ^{15, 16, 17}.”

and Discussion, p. 12:

“In another report, the same group demonstrated CH- π interactions between methyl and carbonyl groups of ubiquitin ¹⁶.”

Reviewer #2 (Remarks to the Author):

Being a non-expert in proteomics, I see the following points the manuscript could potentially be improved. The authors claim their experimental results supported by DFT and MD simulations demonstrated the existence of NH- π bonds.

To me, there are a couple of questions:

(1) The authors appear to believe the observation of J coupling in NMR and Raman should be taken as conclusive evidence that such couplings are mediated by NH- π bonds. Without in-depth knowledge in NMR, I am not sure if this is a unique interpretation. It seems to me that J coupling could be mediated by other mechanisms. I believe it would be appropriate for the authors to make this point clearer.

Response: We thank the reviewer for raising this important point. We agree that the J coupling could in principle be mediated through various non-covalent interactions, e.g., van der Waals interactions as shown in the helium dimer case in the reference mentioned by reviewer 1 in his/her first comment (now added as ref. 42 to the revised manuscript). Our argument for the NH- π bond-mediated J coupling was grounded on the NCI analysis of DFT-based electron density (as shown, e.g., in Fig. 3c), the predicted J couplings (included in the updated Extended Data Tables 1 and 2), and the observation of correlations between the amide proton (of Gly22) and several aromatic carbons (of Phe20), in line with the π character of this interaction. To address the reviewer's comment, we have added the following sentence to Discussion, p. 12:

“While J coupling could in principle be mediated through various non-covalent interactions (e.g., through weak van der Waals interactions⁴²), the DFT calculated electron density, the predicted J couplings, and the observation of J -mediated correlations between the amide proton of Gly22 and several carbons of Phe20's aromatic π ring support the underlying role of the NH- π interaction in mediating the observed magnetization transfer.”

We also exclude the possibility of through-space residual dipolar coupling (RDC) as a magnetization transfer mechanism in this case. Please see below our response to the related comment made by reviewer 3, following which we have added the following sentence to Results, p. 11:

“Possible alternative mechanism of residual dipolar coupling due to molecular magnetic susceptibility anisotropy could be excluded due to its negligibly small size (one to two orders of magnitude smaller than the predicted J of ~ 0.1 - 0.2 Hz)^{39,40}.”

(2) The Raman peak-position based deductions rely on peak position shift by 1 to 3 cm^{-1} . While the spectra presented seem to have very low noise, I wonder if those are the raw data or are those after background subtraction and noise removal. If they make multiple Raman spectra harvesting, a figure including a superposition of say 10 spectra of each sample should be presented. Such a plot would give the readers a sense of the uncertainty in peak position determination.

Response: The Raman spectra reported in Fig. 2(d) have been only subtracted from an almost flat background (polynomial baseline of degree 1 or 2), no other manipulations or spectral processing have been applied. The high signal-to-noise ratio of the profiles has been obtained by accurately setting the experimental conditions (output laser power, high focalization of the beam on the sample, exposition time). In order to check the reproducibility of the measurements and the homogeneity of the sample, we collected, for each peptide, two different spectra by diverse points of the same sample, as shown in Fig. L1. It clearly appears that the two spectra rising from different points of the same sample are practically superimposed, within the spectral resolution (average dispersion of $0.8 \text{ cm}^{-1}/\text{pixel}$). This ensures an uncertainty in the peak position determination of the Raman bands less than 1 cm^{-1} . Figure L1 will be published as part of this Response Letter and hopefully provide the readers with a sense of uncertainty in the determination of Raman peak positions.

Figure L1. Raw Raman spectra of truncated WT (FFAE and FAE) and E22G-A β (FFAG and FAG) peptides reported in the wavenumber range 980-1015 and 1570-1650 cm^{-1} . For each peptide, two different spectra have been collected by diverse points of the same sample in order to check the reproducibility of the measurement (black and red line). The spectra have been only subtracted from an almost flat background, no other manipulations or spectral processing have been applied.

(3) DFT calculations render such a high degree of agreement with the experimental data warrants explanation. This is because experimental results are obtained at room temperature and in the presence of statistical fluctuations such as Brownian motion whereas DFT could only provide the thermal equilibrium configuration at zero degrees kelvin. A discussion about the expected differences is due with an emphasis on the anticipated "proximity to thermodynamic equilibrium configuration" of the peptides under the experimental condition, and thus the anticipated agreement between the DFT results and the experimental results. In other words, how "far" should we expect the configuration of those peptides be from the true equilibrium? An easier to understand way of asking the question would be: If the peptides were left in the analyte for indefinitely long time, would their configuration (secondary, tertiary, etc) be the same as reported in this study? Would they coagulate or precipitate out? If so, would those be the configuration closer to thermal equilibrium than those studied, and shouldn't the DFT study predicts that configuration?

Response: We fully agree that at higher concentrations of this aggregation-prone peptide (A β) the equilibrium state would indeed be an aggregated (fibrillar) form. However, in our study, we used a peptide concentration of $\sim 100 \mu\text{M}$. At this concentration, monomeric A β remained kinetically stable over the timescale of our NMR experiments (typically 7–10 days), and no aggregation was detected. Therefore, the infinite-dilution approximation applied in our MD simulations is justified, and the monomeric ensemble generated by MD reflects a reasonable approximation of the thermodynamic behavior of the monomeric

peptide under our experimental conditions. Importantly, we emphasize that DFT was not applied to arbitrary or artificially optimized structures. Instead, we adopted a hierarchical multiscale approach. First, we performed extensive MD simulations at room temperature using explicit solvent and periodic boundary conditions (PBCs), allowing us to thoroughly sample the conformational landscape of monomeric A β under experimentally relevant conditions. Based on the large distance between the peptide and the edges of the simulation box (minimum distance of 2.45 nm), we confirm that the peptide behaves as an isolated monomer. Following the reviewer's comment, we have now clarified this point by adding this sentence to Results, p. 8:

“The MD ensemble thus generated under the infinite dilution assumption represents a reasonable approximation of the thermodynamic behavior of A β monomers under our experimental condition, in which A β was kinetically stable against aggregation.”

From this MD ensemble, we extracted representative snapshots that reflect thermally accessible conformations. These were clustered based on structural similarity, and the most populated (statistically dominant) cluster's representative snapshots were selected for further interaction analysis via DFT calculations. While DFT itself does not include temperature effects, the input conformations to DFT are derived from an ensemble already accounting for thermal fluctuations and solvent effects, hence reasonably representing room-temperature behavior. It is notable that our DFT calculations were not performed on a single conformation but on multiple representative structures extracted from the dominant MD clusters (total of nine conformers; please see below Fig. L6 in our response to reviewer 3, now included as Extended Data Fig. 10), thereby accounting for ensemble behavior.

We would also like to clarify that our combined MD and DFT study does not aim to provide a full quantitative energetic ranking of conformers but rather to demonstrate the plausibility of this NH- π interaction, which is then verified experimentally by J coupling-based NMR data.

(4) MD simulation requires concrete knowledge of the interaction potentials and for that matter the overall potential energy landscape. That, in the case of individual IDP molecules subjecting to random fluctuations including finite temperature and Brownian motion, and others, should be expected to be a fluctuating factor during the simulation time duration of microseconds. In view of this consideration, the applicability of MD simulation to predicting the experimental observations also warrants further elaboration. In other words, the applicability of MD simulation in this study needs to be rigorously justified.

Response: We wish to stress that our study employed microsecond-scale MD simulations with explicit solvent and periodic boundary conditions (PBCs), ensuring thorough sampling of the conformational space accessible to the IDP system under experimentally relevant conditions. To address the accuracy of interaction potentials and force field applicability-especially for IDPs-, we employed the a99SB-*disp* force field (ref. 31, 10.1073/pnas.1800690115), a model that has been rigorously benchmarked and validated in the literature for both folded and disordered protein systems. Specifically, in this reference, Shaw and coworkers have systematically compared a variety of force fields for their ability to model both ordered and disordered proteins. Their work demonstrated that the a99SB-*disp* provides substantially improved accuracy in simulating IDPs without compromising performance on folded proteins. Importantly, the transferability of a99SB-*disp* across diverse systems supports its use for proteins exhibiting transitions between disordered and ordered states -a property highly relevant for the A β peptide.

While no force field is perfect, a99SB-disp represents a well-supported and physically meaningful compromise within the limits of fixed-charge models. Its success in capturing both local secondary structure propensities and global chain dimensions across a range of disordered systems suggests that it is well suited for our study. We further note that water interactions, which are critical for hydrogen bonding, are explicitly included in our simulations using an optimized water model as per the recommendations in the original benchmark studies. We have further clarified this point by revising the manuscript in Results, p. 8:

“To further investigate the interaction between Gly22 and adjacent aromatic residues, MD simulations of E22G (and WT) A β 40 (each 3 μ s long) were performed using a99SB-disp force field³¹. This force field has been developed to model both ordered and disordered proteins across diverse systems and substantially improved accuracy in simulating IDPs without compromising performance on folded proteins.”

We also acknowledge that MD simulations generate a dynamic ensemble of conformations, and that any given snapshot is inherently transient. To address this, we applied conformational clustering analysis across the MD simulation trajectories. This approach allowed us to identify recurring and statistically significant conformations, which are likely to correspond to narrow local free energy basins (please see below Fig. L5 in our response to reviewer 3, now included as Extended Data Fig. 9). These representative conformers were then subjected to DFT refinement, enabling us to probe specific non-covalent interactions -such as hydrogen bonding and NH- π interactions- at a higher level of theory. In this framework, MD is not used to define a single static structure, but rather to generate a thermally relevant ensemble from which robust structural features can be identified. DFT is then applied to confirm the geometric and energetic plausibility of these features, providing a complementary and rigorous validation of the MD-sampled structures.

In summary, our simulation protocol —based on a state-of-the-art IDP-validated force field, extended sampling timescales, and ensemble-based analysis— provides a robust and scientifically sound foundation for the interpretation of experimental data, particularly in identifying and validating key interactions in disordered peptide systems.

These are just some of the questions that come to my mind after reading the manuscript. I will not be able to provide a substantial opinion about whether the manuscript is suitable for publication in Nature Comm, due largely to my lack of in-depth knowledge pertaining to NMR and also to proteomics.

Response: We sincerely thank the reviewer for presenting some of his/her fundamental concerns and questions. It has led us to further clarify the rationale behind our approach in this Response Letter, which, if accepted, will be published along with the manuscript and SI.

Reviewer #3 (Remarks to the Author):

This work provides a proof-of-principle for the existence of NH- π hydrogen bonds in proteins. The authors studied the NH- π hydrogen bond between the amide proton of Gly22 and the aromatic carbons of Phe20 in the E22G mutant of A β 40. They used a combination of NMR spectroscopy, including 19F labelling, Raman spectroscopy, MD simulations and DFT to provide evidence for the presence of this hydrogen bond.

The critical issue with this work is that a general claim is made based on a single example. It remains unclear whether NH- π hydrogen bonds generally exist in proteins, and if they do, how much they contribute to the conformational stability and other properties.

Response: We thank the reviewer for highlighting this point. The high prevalence of XH- π (including NH- π) interactions in folded proteins, especially in their cores, is based on the established geometric criteria for such interactions and the wealth of structural information for folded proteins accumulated over the past half-century and has been suggested in several studies, as mentioned in the manuscript, pp. 3-4:

“Surveys of protein structures suggest that around 10% of aromatic rings in the compact cores of folded proteins are potentially involved in XH- π (including NH- π) hydrogen bonds^{13,14}.”

On the other hand, the scarce availability of structural information on IDPs does not allow evaluating the prevalence of XH- π interactions in them on the basis of geometric criteria. Another serious challenge in the case of IDPs is their large conformational flexibility and solvent exposure, which makes alternative modes of interactions available to them and hinders reliable prediction of persistent XH- π interactions purely on the basis of geometric criteria. On this background, our study shows that a combined MD/DFT and NMR approach could predict and experimentally verify the presence of such interactions in IDPs (here, in the Alzheimer’s disease-related E22G mutant of A β peptide). Furthermore, our initial bioinformatics analysis, presented in p. 11, shows that the sequence motif Aro-X-Gly or Gly-X-Aro (of which the Phe-Ala-Gly segment of the E22G-A β is an example) is fairly abundant in IDPs/IDRs (around 7%) and suggests that the prevalence of NH- π interactions in IDPs could be higher than what is usually expected. We believe that the experimental high-resolution investigation of XH- π interactions in proteins, especially IDPs, is still at its initial growth phase, and further studies and gradual accumulation of experimental evidence are required to refine the structure- and sequence-based criteria for such interactions in proteins and provide a clearer picture of their general prevalence in proteins, particularly in IDPs. Therefore, we prefer to avoid any general prevalence claim at this premature phase. Instead, we hope that our report triggers the interest in these hitherto overlooked interactions in IDPs and motivates other groups, especially NMR groups, to search for potential “foot traces” of such interactions in IDPs (e.g., anomalous chemical shifts and temperature coefficients). While we fully agree that the general prevalence and impact of XH- π interactions on the stability and other properties of proteins are fascinating aspects of these interactions, we humbly believe that such investigations go beyond the scope of our current study. To address the reviewer’s concern, we have added the following sentence to our Conclusion, p. 14:

“The combined experimental and computational approach presented here promises detection and characterization of such rather overlooked interactions in IDPs, even in systems as extensively studied as A β peptide. Further studies are warranted to clarify the general prevalence and potential impacts of XH- π interactions on the structure, dynamics and function of IDPs.”

To detect the J coupling corresponding to the NH- π hydrogen bond, the authors use a SOFAST-HMQC with ultralong delay approach. This is ingenious and represents state-of-the-art practice for detecting sub-Hz heteronuclear J couplings in biomolecules. However, the measurements are at the technical limit of

NMR sensitivity, making the signal-to-noise borderline to extract conclusions, given possible confounding factors, such as for example residual dipolar couplings or other magnetization transfer mechanisms.

Response: We truly appreciate the reviewer for raising these crucial points. In regard to the signal-to-noise ratio issue, we would like to highlight the following arguments supporting the genuineness of the observed cross-peaks: (i) We have obtained the S/N ratios of ~ 5-7 for cross-peaks between the HN of Gly22 and C γ , C δ and C ϵ of Phe20 (and a smaller S/N ratio of 2 for its C ζ), which are large enough to distinguish the signals of C γ , C δ and C ϵ from the background noise, especially considering that these cross-peaks have been simultaneously observed at the right chemical shifts of these nuclei (see Fig. 4b and its caption). This would be very unlikely if these peaks had their origin in random noise. The cross-peak with C ζ has already been considered a “tentative peak” due to its smaller S/N ratio. (ii) The long-range SOFAST-HMQC measurements of the E22G A β sample were repeated three times, and the same cross peaks were reproducibly observed at the expected chemical shifts. (iii) The long-range SOFAST-HMQC measurement on the wild-type A β sample with the exact same experimental parameters did not lead to any detectable correlation peak (Figure L2). We would also like to mention that the S/N ratios of 5-7 obtained for the cross-peaks of the E22G A β are comparable to the S/N of ~ 3 reported in ref. 15 for the CH $_3$ /pi interaction-based cross-peaks, despite our challenge of working at a much lower protein concentration (0.1 mM A β vs. 2 mM ubiquitin or GB3 in ref. 15) without benefiting from the favorable relaxation properties of methyl groups or the specific isotope labeling scheme used in ref. 15. It is also worth noting that this S/N was achieved after 7-8 days of NMR measurement (vs. 10-20 hours in ref. 15). Further prolongation of the NMR experiment was not possible due to the need for liquid nitrogen refilling of the NMR magnet and the concerns regarding A β sample stability. (iv) The refocusing of J coupling during modified long-range SOFAST-HMQC experiments led to the complete loss of the cross-peaks between the HN and C γ /C δ /C ϵ nuclei (see the next paragraph and Figs. L3 and L4). Taking all these arguments into consideration, we feel completely safe to claim that the observed cross-peaks are genuine signals distinct from the background noise. To address the reviewer’s comment about the signal-to-noise issue, we have now included Fig. L2 in the revised SI as Supplementary Fig. 5 and added the following sentence into Results, p. 10:

“Despite their rather small S:N ratios, these cross-peaks were simultaneously observed at the expected chemical shifts of the involved nuclei, indicating that they are genuine signals. Besides, these cross-peaks were observed in three repeated measurements of freshly prepared E22G-A β 40 samples, further confirming their genuineness.”

The reviewer also presents valid concerns about the underlying mechanism of magnetization transfer in our long-range ^{13}C , ^1H SOFAST-HMQC measurements. To examine the role of J coupling as the mechanism of transfer, we have now performed a new experiment in which we quenched the J coupling-mediated transfer through the addition of 180° proton shaped pulses into the SOFAST-HMQC pulse sequence at the middle of the first and second HMQC delays (see Figure L3). In this way, one could completely refocus the ^1H , ^{13}C J coupling evolution during the first half of HMQC delays and destroy any J coupling-mediated correlations between them. After establishing the performance of this “refocused” SOFAST-HMQC pulse sequence in 1J -based measurement on A β (see Figure L4a), the long-range refocused SOFAST-HMQC measurement was made on the same sample. In line with the underlying role of J coupling as the mechanism of magnetization transfer, the cross-peaks between the HN of Gly22 and the aromatic carbons of Phe20 were completely lost (see Figure L4b,c). These data substantiate the role of J coupling as the magnetization transfer mechanism underlying cross-peaks between the HN of Gly22 and

aromatic carbons of Phe20 in our SOFAST-HMQC spectra. We have now included Fig. L3 in the revised SI as Supplementary Fig. 6 and Fig. L4 in the revised manuscript as Extended Data Fig. 13 and added the following sentence to Results, pp. 10-11:

“Notably, the refocusing of J coupling through the introduction of two additional 180° proton shaped pulses into the long-range SOFAST-HMQC pulse sequence led to the complete loss of these cross-peaks (Extended Data Fig. 13), substantiating the role of J coupling as the magnetization transfer mechanism underlying these correlations.”

In regard to the reviewer’s comment on potential alternative mechanisms of magnetization transfer such as residual dipolar coupling (RDC) in our NMR experiments: in the absence of an external source of alignment that would restrict the rotational motion of molecules and prevent complete cancellation of the spin-spin dipolar coupling in solution, the only source of a non-zero RDC will be the self-alignment of

Figure L2. SOFAST-HMQC (sfHMQC) spectra of the control wild-type A β 40 (WT-A β 40) sample. Top panel: a region of the ^{15}N , ^1H sfHMQC spectrum of the WT-A β 40 containing the HN/N cross-peak of Glu22. Bottom panel: control long-range ^{13}C , ^1H (based on J_{HC} of ~ 0.2 Hz) sfHMQC spectrum of the WT-A β 40 does not show any cross-peak between the HN of Glu22 (at ~ 8.46 ppm, marked by a dashed line) and aromatic carbons of Phe20, nor any cross-peak at the upfield chemical shift of the HN of Gly22 in the E22G-A β 40 (at ~ 7.6 ppm, marked by another dashed line). The contour levels are adjusted to show the background noise (red: positive, green: negative). In the bottom panel, the short-range ^{13}C , ^1H (based on J_{HC} of ~ 165 Hz) sfHMQC spectrum of the WT-A β 40 containing the typical cross-peaks of directly attached ^{13}C and ^1H nuclei in aromatic side chains (in blue) is superimposed for comparison.

Figure L3. Modified ^{13}C , ^1H SOFAST-HMQC (sfHMQC) pulse sequence used for refocusing of J coupling evolution during HMQC delays. Refocusing of J coupling is achieved through introduction of two 180° selective proton pulses (shown in red) at the middle of the HMQC delays (Δ) of the standard sfHMQC pulse sequence. The original 90° and 180° selective proton (light blue) and 90° non-selective ^{13}C (narrow rectangles, black) pulses of the standard sfHMQC pulse sequence are also shown.

molecules in an external magnetic field due to their magnetic susceptibility anisotropy. Indeed, as reported by Bax et al. (10.1006/jmre.1996.1088 and 10.1038/nsb0997-732), the self-alignment of proteins at similar magnetic fields (360-750 MHz) can lead to RDCs of <0.25 Hz for directly attached ^1H - ^{15}N and ^1H - ^{13}C spin pairs (with inter-nuclear distances of 1.02 and 1.09 Angstrom, respectively). Considering that the RDC goes with the inverse cube of the inter-nuclear distance and that the distance between the HN of Gly22 and the aromatic carbons of Phe20 is > 3 times longer than the above-mentioned inter-nuclear distances, the self-alignment RDC between them is expected to be > 27 times smaller and lie below 0.01 Hz. The HMQC delay of 0.25 s used in our long-range SOFAST-HMQC measurements would then lead to $<0.5\%$ evolution of the underlying RDC (compared to the estimated 10% evolution of the through hydrogen-bond J coupling of 0.2 Hz), a negligibly small value that would hardly survive the relaxation losses during the long HMQC delays. Therefore, we can safely exclude RDC as the mechanism underlying the cross-peaks observed between the HN of Gly22 and the aromatic carbons of Phe20 in our SOFAST-HMQC spectra. To address the reviewer's point, we have now added the following sentence to Results, p. 11:

“Possible alternative mechanism of residual dipolar coupling due to molecular magnetic susceptibility anisotropy could be excluded due to its negligibly small size (one to two orders of magnitude smaller than the predicted J of ~ 0.1 - 0.2 Hz)^{39, 40}.”

These technical challenges seem to have guided the choice to use a disordered protein for their proof-of-principle study. However, the existence of conformational disorder introduces problems with the interpretation of the results. It would be important to establish the existence of NH- π hydrogen bonds also in folded proteins.

Response: Thanks for this insightful comment. As stated above, several surveys of protein structure databases have employed the established geometric criteria of XH- π (including NH- π) interactions and suggested that these interactions are prevalent in folded proteins, especially in their compact cores. However, direct experimental verification of them has proven extremely challenging (despite using high

mM protein concentrations, selective isotope labeling schemes, benefitting from favorable relaxation properties of methyl groups etc.) and, to our knowledge, has been successful only in the case of few CH- π interactions (refs. 15 and 16). Our interest in IDPs here is largely motivated by two key points: first, the absence of structural information hinders application of the established geometric criteria to predicting XH- π interactions in IDPs, and second, the access to alternative interaction modes (partly because of conformational disorder, as highlighted by the reviewer, and partly due to higher solvent exposure)

Figure L4. Refocusing of J coupling in the ^{13}C , ^1H SOFAST-HMQC (sfHMQC) experiments. **a**, Short-range ^{13}C , ^1H (based on $^1J_{\text{HC}}$ of ~ 165 Hz) sfHMQC spectra of the E22G-A β 40, obtained without (left) or with (right) J -refocusing pulses (see Supplementary Fig. 6). As expected, introduction of refocusing proton pulses leads to the complete loss of the cross-peaks observed in the absence of them. **b,c**, Long-range ^{13}C , ^1H (based on $^1J_{\text{HC}}$ of ~ 0.2 Hz) sfHMQC spectra of the E22G-A β 40, obtained without (left) or with (right) refocusing pulses. The cross-peaks between the HN of Gly22 and C δ /C ϵ (b) or C γ (c) of Phe20 observed in the standard sfHMQC spectrum (left) are completely lost after introduction of J -refocusing pulses (right), supporting the role of J coupling as the mechanism underlying this correlation. The contour levels in the right spectra of panels a-c are adjusted to show the background noise (purple: positive, green: negative).

influence their existence and stability in IDPs. While the choice of IDPs could in principle bring technical advantages such as more favorable relaxation properties, it is partially compensated in our case by the disadvantage of working with an aggregation-prone IDP (A β peptide) that forced us to measure at much lower protein concentration (0.1 mM) and temperature (hence less favorable relaxation). We fully agree with the reviewer that the single well-ordered structure of folded proteins facilitates the interpretation of experimental results and can provide key additional insights into the nature of these interactions. Indeed, we have already started searching for promising candidates among less challenging folded proteins and potential collaborations with expert groups working on them. At this stage, however, we think that it is an independent project on its own and goes beyond the scope of the current manuscript. We have however included the key insight of the reviewer at the end of our Conclusion, p. 14:

“Moreover, the combined experimental and computational investigation of NH- π interactions in proteins with single well-ordered structures can provide additional key insights into the nature of these interactions.”

The use of the model tripeptide Phe20-Ala21-Gly22 in the DFT calculations is likely to overestimate the interaction energy. Since the study concerns a disordered protein, it is not appropriate to analyze a single configuration of this peptide. DFT results should be reported for multiple configurations.

Response: We appreciate the reviewer’s concern regarding the potential overestimation of interaction energy from DFT calculations based on a single conformation. To clarify, our study does not rely on a single or arbitrarily chosen peptide model; rather, we employed a statistically grounded, multiple conformers approach specifically designed to reflect the dynamic nature of peptides. In response to this comment -and to ensure the robustness and representativeness of our findings-, we have expanded our computational analysis to sample multiple thermally accessible conformations for NCI analysis. This approach explicitly addresses the concern and argues against any overestimation due to a single or arbitrarily chosen peptide structure.

It is notable that to thoroughly explore the potential energy landscape of the A β peptide segment containing the critical NH- π interaction, we performed three independent 3- μ s all-atom MD simulations with explicit solvent and periodic boundary conditions (PBCs). These simulations employed the a99SB-*disp*, which has been specifically developed and validated for modeling IDPs (ref. 31, 10.1073/pnas.1800690115). From these simulations, we extracted three representative snapshots from each trajectory (totaling nine configurations) that exhibited the NH- π interaction between Gly22 and Phe20. Despite the stochastic nature of the MD trajectories, all nine structures exhibited geometrically similar configurations, with an average backbone heavy-atom RMSD of 0.406 Å (Figure L5), indicating that they lie within a narrow free energy basin. This is now included as Extended Data Fig. 9 in the revised manuscript, further supporting the robustness and reproducibility of the observed interaction. We have also added the following sentence to Results, p. 8:

“Superposition of structures extracted from three independent replicates of MD simulations showed similar structures with a low RMSD of 0.406 Å (Extended Data Fig. 9), indicating that they lie within a narrow free energy basin.”

Figure L5. Superimposed structures of peptides extracted from multiple molecular dynamics (MD) simulations showing the Gly22-Phe20 interaction. Structures were obtained from three independent snapshots from each of the three replicates of MD simulations (Run 1 in blue, Run 2 in yellow and Run 3 in green) where the NH- π interaction is observed between Phe20 and Gly22 of E22G variant within the (4 Å, 45°) geometric cutoff. The average RMSD is 0.406 Å for the superimposed structures for E22G variant. The N-H group of interest is highlighted in black circle.

Each of these snapshots was subjected to full DFT-based geometry optimization, followed by non-covalent interaction (NCI) analysis. To characterize the interaction across these conformers, we performed NCI analysis on all DFT-optimized structures. As shown in Figure L6, each structure displayed a similar isosurface (shown in green) between the NH of Gly22 and the aromatic ring of Phe20, consistent with a weak but significant non-covalent NH- π interaction. These NCI plots confirm that the interaction is not an artefact of a single conformation but a recurring feature of the ensemble sampled under experimentally relevant conditions. These results have now been added to the revised manuscript as Extended Data Fig. 10 and mentioned in the text, p. 9, as follows:

“As shown in Fig. 3c (and Extended Data Fig. 10), the light green spikes seen between $-0.01 < \text{sign}(\lambda_2)\rho < 0.01$ (where ρ is electron density and λ_2 is the second eigenvalue of the Hessian of $\rho(\mathbf{r})$ with respect to position, \mathbf{r}) indicate the presence of a weak interaction as a recurring feature in these structures.”

Additionally, to assess the impact of peptide length on the interaction, we performed comparative DFT and NCI analyses on both the tripeptide (Phe20-Ala21-Gly22) and an extended pentapeptide fragment (including adjacent residues from the E22G variant). As shown in Figure L7, the NH- π interaction persists in the pentapeptide model, confirming that it is not artificially stabilized by truncation. Due to the high computational cost of DFT on longer peptides and the consistent interaction profile observed, further extension was deemed unnecessary.

Finally, we wish to stress that our study does not aim to report absolute interaction energies or rank conformational states. Rather, the DFT calculations were used to validate the geometric and qualitative energetic plausibility of NH- π interactions identified through MD simulations. The convergent behavior across multiple independent configurations and different peptide model sizes reinforces the general validity of our findings and argues against any overestimation due to a single or arbitrarily chosen model for peptide.

Figure L6. Non-covalent interaction (NCI) analysis of the truncated peptide structures showing the Gly22-Phe20 interaction in the E22G variant. NCI analysis performed on truncated capped peptide structures obtained from three independent snapshots from each of the three replicates of MD simulations where the NH- π interaction is observed between Phe20 and Gly22 of E22G variant within the (4 Å, 45°) cutoff shown here. The computed electron density distribution (ρ) of the different conformations of the truncated peptide structures and their respective reduced density gradient (RDG, s) versus $sign(\lambda_2)\rho$ are shown (color scheme: blue, attractive interactions; green, weak interactions; red, repulsive interactions). The RDG isosurface is at $s = 0.3$ a.u.

Figure L7. NCI region between N-H of Gly 22 and the phenyl ring of Phe20 in a pentapeptide remain consistent to previous analysis for tripeptide despite of C and N terminal extension of peptide. The computed electron density distribution (ρ) of the truncated capped structure of the pentapeptide and its reduced density gradient (RDG, s) versus $\text{sign}(\lambda_2)\rho$ are shown (color scheme: blue, attractive interactions; green, weak interactions; red, repulsive interactions). The RDG isosurface is at $s = 0.3$ a.u.

We thank the three reviewers of our manuscript for their positive evaluation of our first revision (reviewers 1 and 3) and further useful comments (reviewers 2 and 3). Below, please find our point-to-point response to the comments made by them.

Reviewer #1 (Remarks to the Author):

The authors have taken all reviewers' comments seriously and addressed them thoroughly. Additional computational and experimental data have been added to solidify the claims made in the paper. Claims have been adjusted appropriately and are supported by the data. It is true that this still only represents one experimental example as pointed out by reviewer 3, but the authors have done a good job in addressing this point.

Overall the work is sound and conclusions are not overstated.

Response: We appreciate the reviewer for his/her remarks and are happy to see that (s)he is satisfied with the revised manuscript.

Reviewer #2 (Remarks to the Author):

I have to say that the response of the authors left still quite a few questions unanswered. Below I list my remaining questions.

Response: We appreciate the reviewer's insightful questions and try our best to address them below.

(1) The authors cited once again the reasons they think the observed J-coupling should be due to NH-pi bonds. What would be helpful is if they provided arguments why other cause for J-coupling being unlikely. They seem to believe DFT calculation to be strong supporting evidence without a satisfactory answer to my question in (3) questioning the validity of the accuracy (see below for more details).

Response: Our argument for the presence of an NH-pi bond proceeds as follows:

a, the observed correlation between the amide 1H of Gly22 and aromatic 13Cs of Phe20 is mediated, not directly through-space (via mechanisms such as residual dipolar coupling, which was ruled out), but indirectly through-electron, i.e. it is a J coupling-based correlation,

b, several lines of experimental evidence (e.g. upfield NMR chemical shift for amide 1H and 15N of Gly22, 19F chemical shift perturbation for Phe20, the loss of upfield chemical shift after fluorination of Phe residues, 1H,1H NOE correlation between Gly22's HN and Phe20's Hε) confirm the spatial proximity of Gly22 and Phe20's ring,

c, the near-zero temperature coefficient of Gly22's upfield chemical shifts, supported by slight rigidification of Phe20 suggested by 19F R1 data and changes in the vibrational frequency of Phe's rings in the Raman spectroscopy data, point to the presence of an interaction stabilizing this spatial proximity, despite the high conformational flexibility of the intrinsically disordered Aβ peptide and temperature changes,

d, the MD simulation shows that the E22G-Aβ40 (unlike the WT-Aβ40) populates conformations, which according to the well-established geometric criteria in this field, contain an NH-pi interaction between the

Gly22's HN and Phe20's ring. It is worth noting that the MD force field does not contain a specific term representing such XH- π interactions in proteins, nevertheless, a significant fraction of the E22G-A β 40 conformers in the MD trajectory satisfy all three geometric criteria of this interaction in a persisting manner.

e, while the presence of this NH- π interaction is at this point established by our MD data, further insights on this interaction are provided by DFT calculations, through 1) NCI analysis showing spikes of zero density gradients at small density values, which are spatially spread over the π ring, and 2) DFT prediction of the J-based correlations, indicating the capability of this interaction to serve as an electron density-based path for the J-based correlations mentioned in section a above.

Below in sections 3 and 4, we will reply to the reviewer's comment on the validity and accuracy of our DFT and MD data, but we would like to (re)draw the reviewer's attention to this point that the MD data combined with the application of geometric criteria is sufficient to establish the presence of the NH- π interaction in our study. The role of DFT data is to provide insights on the nature of this interaction and generate testable predictions such as the mentioned J couplings.

(2) Comparison of two consecutive measurements of FFAE, for example, leads to noticeable questions regarding the assurance provided by the authors. The signal-to-noise ratio of the red and black spectra are obviously different. However, in the context of how the authors used Raman spectra to justify the trustworthiness of their DFT results pertains to the Raman peak position shift of 1-3 cm^{-1} , they need to provide say 10 superimposed Raman spectra to provide a quantitative reproducibility of the peak positions including the wavenumber increment size. The information would help the readers to judge the certainty when comparing DFT with Raman measurements and thus the significance of the 1-3 cm^{-1} peak position shifts and is in my opinion absolutely necessary.

Response: Thanks for the reviewer's comment. As recommended, we have now carried out an additional set of Raman measurements on the peptide samples (see Figure L1). The visible Raman spectra were collected in the wavenumber range 400-4000 cm^{-1} using XploRA PLUS Confocal Raman microscope equipped with a laser emitting at 633 nm. The Raman signal was collected in back-scattering geometry and detected with a charge-coupled device (CCD) detector. A grating element with 1800 grooves/mm and a slit width of 50 μm was employed throughout the experiments to have a resolution of about 1.2 cm^{-1} . The aqueous solutions of tripeptides FAE and FAG were prepared at the same concentration used for the previous Raman experiments and deposited on a glass slide, waiting for few minutes until water molecules were mostly evaporated. The excitation wavelength was focused onto the sample with a laser spot area of about 0.85 μm^2 through a 50X objective (NA ~ 0.75). The Raman measurements were performed several times (at least 10 points for each sample) to ensure the reproducibility of spectra for each sample. All experimental parameters were implemented in a manner to ensure that any potential photodegradation effects were avoided. The 10 superimposed spectra shown in Fig. L1 clearly confirm the reproducibility of the wavenumber position of the main Raman bands of interest for the two peptides. Moreover, the shift in phenyl stretching at around 1608-1610 cm^{-1} between samples FAG and FAE are confirmed, within the experimental resolution, and in agreement with the results shown in the manuscript (Extended Data Fig. 5). Similarly, a high level of reproducibility was observed for the FFAE peptide (Fig. L1e,f, unfortunately, due to sample availability limitation, the FFAG peptide could not be measured). Considering that these data were collected using different samples (for the previous and new measurements), on multiple spots of the same samples

(two spots in the previous measurements and 10 spots in the new measurements) and in two different labs, we feel safe to defend the reproducibility of our Raman data. **Figure L1a-d** is now included as a new **Supplementary Fig. 4** in the revised SI and referred to in the manuscript, page 7.

Figure L1. Ten superimposed Raman spectra of the truncated WT (FAE, top; FFAE, bottom) and E22G mutant (FAG, middle) tri- or tetra-peptides collected from ten different points of peptide samples, supporting the high reproducibility of the wavenumber positions and shifts. In addition, the wavenumber positions and shifts match (within the experimental resolution) those shown in the Extended Data Fig. 5, collected from two different points of another set of FAE/FAG/FFAE samples. Due to sample availability limitations, these double-check measurements were not performed for the FFAG peptide.

We would also like to add the following technical points about our previous Raman measurements: as known, the reproducibility of the peak positions in Raman spectra depends on 1) technical factors of the Raman instrument like spectral dispersion, laser stability and repeatability of the detector correction and 2) on the features of the sample, such as homogeneity. Concerning the last issue, the spectra reported in the previous response letter (now included as Fig. L2) demonstrated the homogeneity of the sample, within the experimental resolution. The S/N ratio of the two spectra collected from different points of the spot sample was different, as expected, due to the diverse volume of scattering of the two measurements. However, the

good overlap of the spectra in Fig. L2, within the experimental resolution, confirmed the homogeneity of the sample.

Figure L2. Raw Raman spectra of truncated WT (FFAE and FAE) and E22G-A β (FFAG and FAG) peptides reported in the wavenumber range 980-1015 and 1570-1650 cm^{-1} . For each peptide, two different spectra have been collected by diverse points of the same sample in order to check the reproducibility of the measurement (black and red line). The spectra have been only subtracted from an almost flat background, no other manipulations or spectral processing have been applied.

As specified in the experimental methods section, we had collected the Raman spectra using an integrated micro-Raman spectrometer Horiba Jobin Yvon, model LabRam Aramis, with an average dispersion in the range 400-4000 cm^{-1} set at 0.8 $\text{cm}^{-1}/\text{pixel}$. In Fig. L3 we report the curve of the dispersion of the Raman spectrometer for the 1800 grooves/mm grating as a function of the wavelength. The spectral range 1000-1610 cm^{-1} where we observe the Raman bands of the peptides (Extended Data Fig. 5 in the manuscript) corresponds to a wavelength range 561-582 nm. In this window the dispersion is less than 0.5 $\text{cm}^{-1}/\text{pixel}$, as detectable by looking at the plot in Fig. L3. The technical sheet of LabRam Aramis indicates that, under normal conditions of temperature stability ($\pm 1^\circ \text{C}$), the repeatability is 1 pixel with a standard CCD. This means that a maximum variation of $\pm 0.5 \text{ cm}^{-1}$ can be observed in the peak position of our Raman spectra, ensuring a quantitative reproducibility consistent with the shift of 1-3 cm^{-1} we discuss in the manuscript. Moreover, we have checked that in the same working conditions of temperature and humidity the wavenumber center of the laser emission is absolutely stable during the time within the experimental resolution of the spectrometer.

Figure L3. Dispersion of the Raman spectrometer as a function of the wavelength for the 1800 grooves/mm grating; the pink curve represents the element pixel size of CCD in cm^{-1} .

In regard to the comparison between the experimental and DFT calculated Raman spectra, we would like to mention that it is a common practice to use the DFT calculated spectra for assigning the experimental bands to specific vibrational modes and interpreting the spectral changes detected through experiment. Since the DFT calculations typically tend to overestimate the vibrational frequencies (sometimes, appropriate scaling factors are used to correct the calculated frequencies, see e.g. 10.3390/molecules26164790 and 10.1002/prot.24229), the aim of theoretical calculations is often limited to detect the trends and evaluate the qualitative agreement with the experiment (see e.g. 10.3390/foods12152888 and 10.1002/jrs.2150). Following this common practice in the field, we used DFT calculations carried out on the truncated A β peptides in order to verify that the Glu \rightarrow Gly substitution induces a shift in the frequency position of the ν_{8a} vibrational mode of the Phe's ring, in qualitative agreement with our experimental observations. This point has been added to the Methods section, page 24:

“The DFT-predicted and experimental Raman spectra of the truncated WT (FAE) and mutated (FAG) tripeptides were compared in order to assess the qualitative agreement between the DFT and experiment regarding mutation-induced Raman band shifts.”

(3) The authors seem to have misunderstood my question about whether one should expect DFT calculations to render that degree of agreement with the experimental results given they are supposedly accurate for samples under totally different temperatures (300 K vs 0 K). In other words, at the core of the question is the role played by entropy. The argument they provided talks about the kinetics instead of thermodynamics of the sample system. One case in point is that just because a piece of glass does not incur significant change over a time period of say 10 years does not in any way infer the proximity of the glass to thermodynamic equilibrium, i.e. the single crystalline state of SiO_2 , which is the state relevant to DFT calculations.

Response: We appreciate the reviewer's clarification regarding the role of entropy and the distinction between thermodynamic and kinetic stability. We have now revised the manuscript to make the following points explicit:

1. Thermodynamic equilibrium of A β 40 monomers: At sub-critical concentrations for aggregation (<0.1–1 μ M), the thermodynamic equilibrium state of A β 40 is the monomeric peptide. Our MD simulations are carried out under infinite dilution, such that the simulated ensemble represents the thermodynamic equilibrium state of monomeric A β 40. The DFT calculations are therefore applied to structures relevant to the monomeric equilibrium state.
2. Experimental conditions of NMR measurements: Due to sensitivity limitations of high-resolution NMR, the peptide had to be studied at higher concentrations (50–100 μ M) and over prolonged acquisition times (~8 days for J-coupling experiments). At these concentrations, the monomeric state is not the global thermodynamic equilibrium but rather a kinetically stable form of the peptide. Careful sample preparation (removal of pre-formed aggregates) ensured that the system remained in this kinetically trapped monomeric state during measurement.
3. Role of MD and DFT in our workflow: Entropy and conformational heterogeneity are explicitly treated at the MD stage through explicit-solvent sampling, ensemble statistics, and clustering. DFT is then applied not to predict the global thermodynamic minimum, but to refine the local geometry and enthalpic strength of specific NH– π contacts within thermally populated snapshots. Thus, DFT serves as a local probe, not as a predictor of global equilibrium.
4. Agreement between computation and experiment: we recognize the reviewer's important point—agreement between DFT-refined MD conformations and NMR observables is not guaranteed in principle, as clearly illustrated by the glass analogy. However, in our case, the consistency arises because:
 - o MD provides a finite-temperature ensemble that reflects the thermodynamic equilibrium of monomeric A β 40 at infinite dilution.
 - o The experimental J couplings are primarily sensitive to local structural features, which are preserved in the kinetically stable monomer samples used for NMR.
 - o DFT validation of NH– π interactions ensures that these motifs are enthalpically plausible within the MD ensemble.

We have added these points to the revised manuscript in several places in pages 8 and 20-23.

(4) My final question was directed at the validity of the potential or the force field as the authors call it, used in their MD simulation, and the time variability of the force field. I must once again make it clear about my lack of specific knowledge background of the field of sciences in discussion. My question stems from the consideration of the molecular scale environment of IDP molecules in solution. Given that the molecule-molecule interaction becomes significant only when two molecules are in close proximity (~2.5 nm as pointed out by the authors) to one another, and that at this scale the environment and thus the potential or the force field is expected to fluctuate significantly with time, I believe the authors are obligated to provide convincing arguments why the force field they employed is valid not pertaining to the experiments in the cited references but to the formation and dissociation of NH- π bonds.

Response: We thank the reviewer for this thoughtful question. We clarify our points as follows:

Our reference to 2.5 nm was intended to indicate that at this separation the simulation mimics infinite dilution conditions, thereby excluding intermolecular peptide–peptide interactions. Under these conditions, the environment of the A β monomer is defined solely by explicit solvent molecules and intramolecular flexibility, without fluctuation from peptide–peptide contacts. We wish to stress that dynamic variability is intrinsic to the MD description. Although the force field functional form is time-independent, MD trajectories are time-dependent. The potential energy landscape is recalculated every time step based on instantaneous atomic coordinates, so the observed variability comes from the thermal motions of atoms in the MD trajectory, not from changes in the force field itself. Thus, Brownian/thermal fluctuations and proximity-dependent interactions are inherently included in the simulation trajectories.

For our MD simulation, we employed the a99SB-disp force field with its matched water model, which is state-of-the-art for intrinsically disordered proteins (IDPs) (#REF:10.1073/pnas.1800690115). It was specifically optimized to reproduce both secondary-structure propensities and global conformational heterogeneity of disordered peptides. Although it does not contain a specific parameter for NH– π interactions, the MD trajectories sampled conformations that persistently satisfy the known geometric requirements for such contacts. This demonstrates that the force field captures these motifs in practice, even without an explicit term. The transient nature of NH– π contacts is treated statistically: we analyze microsecond-scale trajectories via clustering and contact statistics, reporting recurrent geometries, lifetimes, and occupancies. Thus, our results represent features of the ensemble free-energy landscape, not isolated short-lived encounters. To ensure accuracy in the short-range noncovalent physics, we carried out DFT calculations on representative MD snapshots. This multiscale workflow provides a direct test of NH– π interaction stability, addressing the reviewer’s request for validation “pertaining to the formation and dissociation of NH– π bonds,” rather than relying solely on force field parametrization. Finally, our NMR J-coupling data independently corroborate the presence of the proposed NH– π interaction in solution. This serves as an orthogonal validation of both the sampling provided by the force field and the local interaction model.

We have added these points to the revised manuscript in several places in pages 8 and 20-23.

While I did not want to review the rebuttal, the reply of the authors gave me an uneasy feeling about the substance of this work and thus these follow up questions.

Response: We thank the reviewer for sharing his/her concerns, which have helped clarifying some fundamental aspects of our work. We hope that our arguments are now sufficiently clear and the reviewer is satisfied with them.

Reviewer #3 (Remarks to the Author):

The present work now convincingly documents one NH– π case in an IDP. While the general claims are now tempered, and the challenge and independent scope of studies on folded proteins are explicitly acknowledged, there remains no experimental evidence of NH- π bonds in folded proteins, where interpretation would be less confounded by conformational heterogeneity.

The Title, Abstract, and Conclusions should unambiguously state that this proof-of-principle is for an

intrinsically disordered system and avoid implying general prevalence in folded proteins beyond citing structure-survey expectations. It may help to add one sentence to the Abstract noting that validation in folded proteins remains future work.

Response: We thank the reviewer for his/her positive evaluation of our revision. We fully agree with the reviewer that our work presents as a proof-of-principle one case of an NH- π hydrogen bond in an IDP. We also share this opinion with the reviewer that future studies are required to search for more cases of NH- π bonds in proteins, especially in folded proteins where the interpretation of experimental data could be less challenging than IDPs. We believe that the current Title, “Direct detection of an NH- π hydrogen bond in an intrinsically disordered peptide”, is explicit about the restricted scope of our work (an NH- π case in an IDP). To further highlight the reviewer’s point, and following his/her recommendation, we have modified the Abstract, as follows:

“Our results present a proof-of-principle example of NH- π interactions in an intrinsically disordered protein (IDP) and suggest the potential prevalence of π hydrogen bonds on the surface of IDPs. Direct experimental verification of NH- π interactions in folded proteins remains for future studies.”

added the following sentence to Discussion, page 12,

“In this study, we report as a proof-of-principle one case of NH- π interactions found on the surface of an intrinsically disordered peptide, namely the E22G A β peptide, ...”.

and the following ones to Conclusion, page 14:

“The reported XH- π interaction serves as a proof-of-principle example in IDPs, while direct experimental verification of NH- π interactions in folded proteins remains for future works. Further studies are warranted to clarify the general prevalence and potential impacts of XH- π interactions on the structure, dynamics and function of IDPs. Moreover, the combined experimental and computational investigation of NH- π interactions in proteins with single well-ordered structures can provide additional key insights into the nature of these interactions beyond the current structure-based surveys.”

We hope that these additions ensure conveying this key point to future readers of this manuscript and satisfy the esteemed reviewer.